# Effectiveness Assessment of an Innovative Ejector Plant for Port Sediment Management

**Marco Pellegrini** [1,2], **Arash Aghakhani** [1], **Maria Gabriella Gaeta** [2,3], **Renata Archetti** [2,3,*],
**Alessandro Guzzini** [2] **and Cesare Saccani** [1,2]

1    Department of Industrial Engineering, University of Bologna, 40136 Bologna, Italy;
     marco.pellegrini3@unibo.it (M.P.); arash.aghakhani2@unibo.it (A.A.); cesare.saccani@unibo.it (C.S.)
2    Interdepartmental Centre for Industrial Research in Building and Construction, 40131 Bologna, Italy;
     g.gaeta@unibo.it (M.G.G.); alessandro.guzzini2@unibo.it (A.G.)
3    Department of Civil, Environmental, Chemical and Materials Engineering, University of Bologna,
     40136 Bologna, Italy
*    Correspondence: renata.archetti@unibo.it

**Abstract:** The need to remove deposited material from water basins is common and has been shared by many ports and channels since the earliest settlements along coasts and rivers. Dredging, the most widely used method to remove sediment deposits, is a reliable and wide-spread technology. Nevertheless, dredging is only able to restore the desired water depth but without any kind of impact on the causes of sedimentation and so it cannot guarantee navigability over time. Moreover, dredging operations have relevant environmental and economic issues. Therefore, there is a growing market demand for alternatives to sustainable technologies to dredging able to preserve navigability. This paper aims to evaluate the effectiveness of guaranteeing a minimum water depth over time at the port entrance at Marina of Cervia (Italy), wherein the first industrial scale ejector demo plant has been installed and operated from June 2019. The demo plant was designed to continuously remove the sediment that naturally settles in a certain area through the operation of the ejectors, which are submersible jet pumps. This paper focuses on a three-year analysis of bathymetries realized at the port inlet before and after ejector demo plant installation and correlates the bathymetric data with metocean data (waves and sea water level) collected in the same period. In particular, this paper analyses the relation between sea depth and sediment volume variation at the port inlet with ejector demo plant operation regimes. Results show that in the period from January to April 2020, which was also the period of full load operation of the demo plant, the water depth in the area of influence of the ejectors increased by 0.72 mm/day, while in the whole port inlet area a decrease of 0.95 mm/day was observed. Furthermore, in the same period of operation, the ejector demo plant's impact on volume variation was estimated in a range of 245–750 m$^3$.

**Keywords:** ejectors; sediment transport rate; port sediment management; effectiveness assessment

## 1. Introduction

The presence of anthropic activity in the coastal environment strongly modifies waves, currents and sediment transport regimes. In particular, intense wave-induced currents and sediment transport rates are present around ports and commonly influence their commercial and recreational activities. The accumulated sediments reduce the admitted draft of the navigation channel on the one hand and generate erosional effects on the leeside coasts on the other. As a consequence, harbours frequently require ordinary maintenance dredging to remove the accumulated sediments [1]. Dredging is a consolidated and proven technology, but involves considerable drawbacks [2–5], since it has a notable environmental impact on the marine environment, contributes to the mobility and diffusion of contaminants and pollutants already present in the settled sediments, and obstructs navigation during its operational phases. Periodic hydrographic surveys of the harbour

area are essential for the accurate determination of quantities and timing of maintenance dredging. Since maintenance dredging is often performed on an as-needed basis, regular surveys become an essential tool to properly time the work [6]. Furthermore, bathymetries are surveyed before and after dredging operations to estimate the volume of the sediment removed. On the other hand, the use of dredging equipment allows the measuring of the material dredged, which is usually defined by contract.

New approaches have been developed over the years as alternative methods to dredging. In particular, [7] classifies alternative solutions to dredging in three categories: (i) anti-sedimentation structures, (ii) remobilizing sediment systems, and (iii) sand by-passing plants. Anti-sedimentation structures considerably reduce the amount of sediment to be removed from harbour inlets but present environmental concerns and still require sediment removal [8]. Remobilizing sediment systems use an injection of water or the movement of mechanical devices such as dredger propellers to cause the lift and the separation of the grains from the seabed. In particular, water injection dredging has been widely applied as a cheaper and less impacting solution than traditional dredging [9]. Nevertheless, environmental issues due to the lack of control of the resuspended sediment deserve further investigation [10,11], while some technical limitations are present if the sediment is mainly composed of sand. Sand by-passing plants have limited environmental impacts but are characterized by relatively high installation costs and often uncertain operational costs.

An innovative sand by-passing technology tested in the framework of the LIFE MARI-NAPLAN PLUS and STIMARE projects [12–14] is based on a patented undersea ejector able to keep the designed draft of the entrance channel over time through a continuous removal of settling sediments. If the sediment is properly handled, the instantaneous removal can eventually also produce benefits with regard to counteracting neighbouring erosion processes. The ejector (Figure 1) is an open jet pump (i.e., without a closed suction chamber and mixing throat) with a converging section instead of a diffuser and a series of nozzles positioned circularly around the ejector. The ejector has a diameter of about 250 mm and a whole length of about 400 mm. Each ejector is placed on the waterbed and transfers momentum from a high-speed primary water jet flow to a secondary flow that is a mixture of water and the surrounding sediment. The sediment–water mixture is then conveyed through a pipeline and discharged in an area where the sediment can be picked up again from the main water current or where it is not an obstacle for navigation. Both water feeding and discharge pipelines are DN80 spiral tubes (external diameter of about 90 mm). Based on a preliminary test carried out in 2017 [14], it is known that with a primary water feeding flowrate of about 27 $m^3$/h, a working pressure of about 2.4 bar, and a discharge pipeline characterized by a 60 m length, one ejector is able to convey a peak sand flowrate at the discharge pipeline of about 2 $m^3$/h with a water pump power consumption of about 3.5 kW. In the same peak working condition, the whole discharge flowrate of one ejector is about 34 $m^3$/h of water–sediment mixture, or a peak sediment concentration of about 6% in volume. Each ejector works on a limited circular area created by the pressurized water outgoing from the central and circular nozzles, whose diameters depend on the sediment characteristics such as, for example, the repose angle. By ejector integration in series and in parallel, it is possible to create or to maintain a seaway.

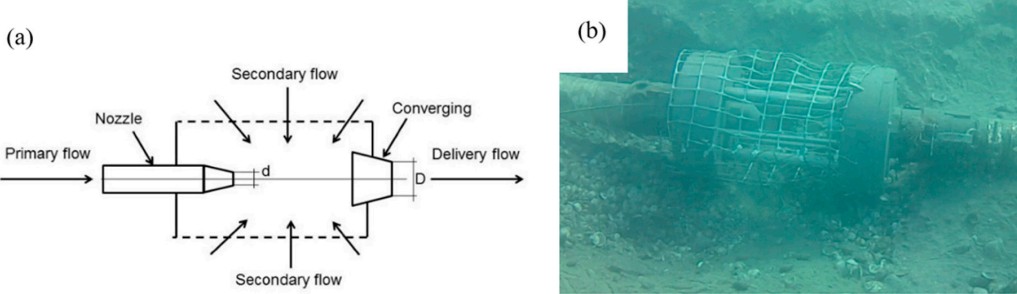

**Figure 1.** (**a**) Sketch of the ejector [14] and (**b**) underwater picture of the ejector in operation.

The technology is reliable since, generally speaking, jet pumps have been applied since the 1970s for coastal applications. The ejector technology has been developed and tested starting in 2001 by the University of Bologna and the start-up Plant Engineering Srl. In 2005, the first experimental plant was realized and tested in the port of Riccione (Italy). In 2012, a second experimental plant [15,16] was realized in Portoverde Marina (Italy). Both installations have been realized at port entrances and were designed to handle sand. A third experimental installation was realized in 2018 in Cattolica (Italy), wherein for the first time the ejectors were applied in the management of silt and clay sediments and installed in a river channel [17].

An ejector demo plant has been realized at the port entrance of the Marina of Cervia (Italy) and has been in operation from June 2019 to September 2020.

The aim of the paper is to assess the ejector demo plant effectiveness, which can be defined as the ability to be successful in maintaining the navigability at the port entrance over time. The novelty of the study is related to the methodology applied for the evaluation of the impact of the ejector demo plant on both water depth and sediment volume variations at the port inlet. In fact, for dredging and other alternative technologies that operate over a short time, the evaluation of the impact is based on the comparison of bathymetries of the interested area before and after sediment removal; the ejector plant works continuously and so the effects are monitored over a long period. Therefore, natural sediment transport is also relevant and should be taken into account in the effectiveness evaluation. The prediction of sediment transport is very complex: it always needs an accurate phase of calibration and validation based on measurements and it requires a good sediment transport model driven by reliable input data of waves and currents, initial bathymetry, sediment characteristics, etc. A large amount of literature is available and interesting examples can be found [18–22]. Nevertheless, the analysis of effectiveness of a continuous working sand by-passing system has never been approached, and in previous applications the by-passed amount of sediment has been always estimated starting from dredging needs [23]. This paper's investigation was carried out by making a comparison of water depth and sediment volume variation over time before and after ejector demo plant installation through the analysis of bathymetries and metocean data. Therefore, the sediment transport rate in the area of Cervia port entrance was firstly assessed, starting from the analysis of the detailed bathymetries realized in the last 3 years. Moreover, the 3-year metocean climate on the site was analysed and discussed together with the bathymetric evidence. Then, the operation period of the ejector demo plant was compared with the two previous years, from June 2017 to June 2019. The effectiveness of the demo plant was evaluated on the basis of the different operation and control strategies tested.

## 2. Description of the Ejector Demo Plant

### 2.1. Description of the Study Site: Cervia, North Italy

The harbour of Cervia (Figure 2) is located in the coast of Emilia-Romagna region, Italy. It was designed and realized along an artificial canal to convey the salt produced in the near salt flats. Therefore, the harbour had a very important role in the past, interconnecting the land and the maritime markets. Maintenance dredging activities have been used since the first half of the nineteenth century, when docks started to be lengthened to balance coast advancement. A new design of harbours occurred during the 1970s when the local municipality decided to modernize the existing infrastructure and to realize a marina able to satisfy users' demands.

The Marina of Cervia currently extends over an area of approximately 43,000 m$^2$ with a capacity of around 300 berths. A further lengthening of the docks (20 m for the southern dock and 40 m for the northern dock) was planned and realized in 2009 by the municipality as a countermeasure against the coast advancement and to prevent port inlet sedimentation.

Nevertheless, traditional dredging and sediment handling through dredger propellers (the second being a remobilizing sediment technique in which dredger propellers are used

to remobilize the sediment from the seabed) were still planned and periodically operated at the port entrance [14]. In the period 2009–2015, more than 17,000 m$^3$ of sediment was removed yearly, with dredging from the basin being responsible for a total expenditure of about EUR 1 million—i.e., a weighted average cost of EUR 8.31 per dredged cubic meter of sediment. Furthermore, in the same period, the Municipality of Cervia invested about EUR 350,000 in propeller operations, almost once per year. The physical-chemical characterization of the sediment present at the port inlet revealed that the main component is sand (97%), while the specific weight is 1.9 g/mL.

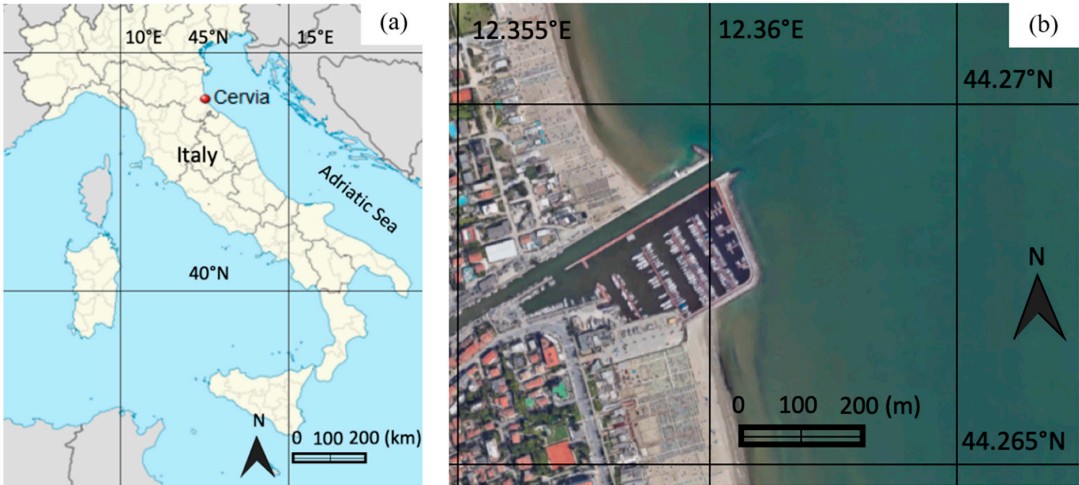

**Figure 2.** Cervia position (**a**) and harbour aerial picture of the study area (**b**).

The annual net longshore sediment transport in the area offshore the port of Cervia is estimated to be equal to zero [24]. Therefore, the Cervia port area is commonly defined as a convergence point for the annual longshore sediment transport, with the convergence point position being affected by annual wave climate. As it is common in all the Northern Adriatic Sea, the wave climate is characterized by severe storms mainly generated by north-easterly winds, named Bora, even if south-easterly winds, named Sirocco, may have relevant seasonal impacts [25,26], with the latter generally inducing the highest surge levels [27]. Details on the wave buoy are available at [28]. The wave roses in Figure 3 show the annual distribution of the significant wave height (left panel) and the peak period (right panel) versus the mean wave directions at the buoy showing: (i) the most energetic waves, up to 4.0 m in height, propagating from the sector 50–60° N; (ii) the most frequent conditions, with wave heights up to 1.0 m, coming from 90° N; (iii) the high-wave periods with values ranging from 9 to 11 s, coming from 90° N; (iv) the most frequent values range from 5 to 7 s.

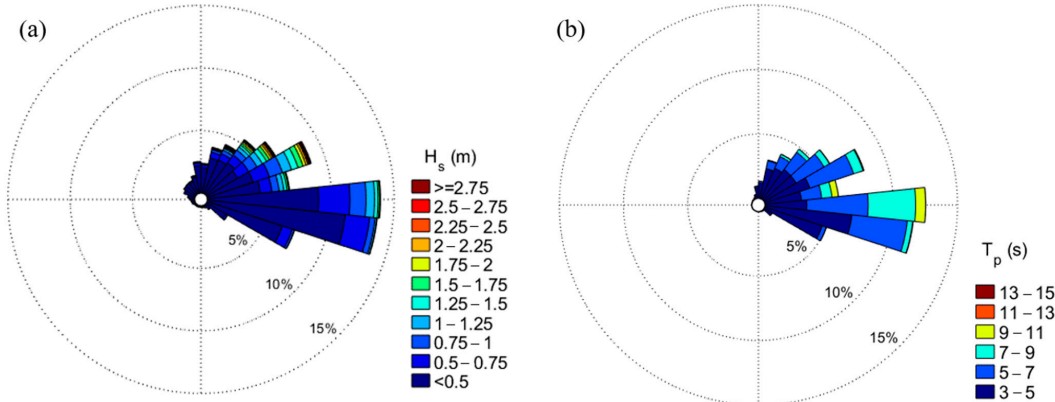

**Figure 3.** Wave rose from the Nausicaa buoy in the period 2010–2020: polar distribution of the significant wave height (**a**) and peak period (**b**).

### 2.2. The Ejector Demo Plant of Cervia

As a possible alternative solution of the port sedimentation problem, the ejector technology was proposed and tested starting from 13 June of 2019. The ejector demo plant installed in Cervia has the main objective of guaranteeing navigability at the port inlet while in operation. The Cervia demo plant consists of 10 ejectors located at the port entrance, as shown in Figure 4, with in- and out-flow pipelines laying on the seabed, delivering the mixture discharge composed of the moved sediments and water in a location south of the port entrance channel.

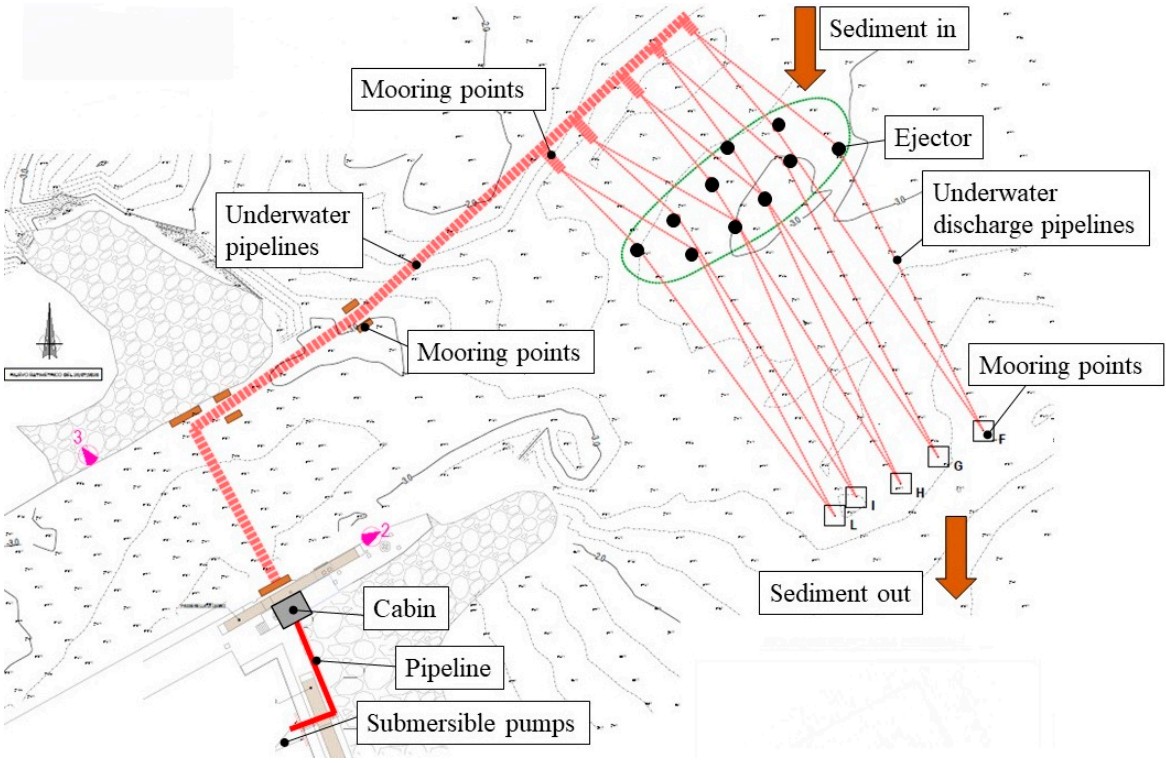

**Figure 4.** Location of ejectors in the demo plant of Cervia.

The Cervia demo plant also includes a fully automated and remotely accessible pumping station equipped with auto-purging filters. The Piping and Instrumentation Diagram (P & ID) of the pumping plant is schematically shown in Figure 5, where only one ejector line is drafted. There are two submersible pumps, each one feeding five ejectors. Each pumping line has an auto-purging disk filter: the auto-purging cycle is activated once the pressure drop in the filter reaches a certain level. The total pumped water flowrate is controlled by an inverter, while the flowrate for each ejector feeding pipeline is balanced through electrovalves. An air compressor can be used to inject compressed air in the line to easily identify the positions of the ejectors on the seabed. The total installed power is about 80 kW. A local meteorological station measuring both wind speed and direction has been installed to relate plant operation with sea weather conditions—in particular, when wind speed overcomes a predefined threshold, which indicates the risk of heavy sea, the water flowrate feeding the ejectors is set at the maximum value (about 30 $m^3$/h per ejector) to guarantee a sufficient sediment suction and conveying capacity.

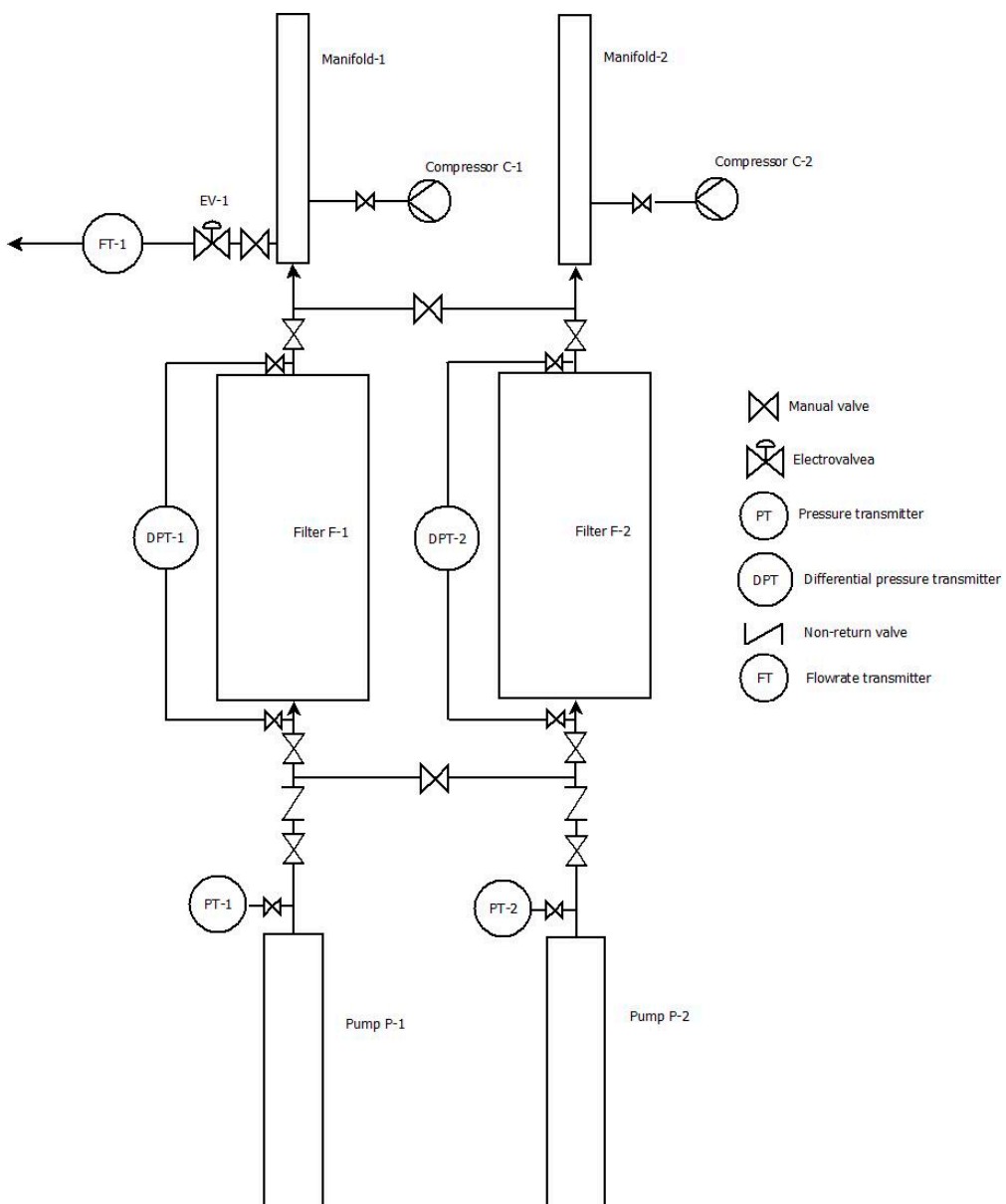

**Figure 5.** Piping and Instrumentation Diagram (P & ID) of the pumping plant [13].

## 3. Materials and Methods

### 3.1. Ejector Demo Plant Operation and Monitoring

Cervia's ejector demo plant was operated continuously from June 2019 to September 2020, thus achieving the objective of the LIFE MARINAPLAN PLUS project—namely, the monitoring of performance and impacts produced [14] for a minimum period of operation of 15 months. Figure 6 summarizes the five operating phases in which ejector demo plant operation can be divided. In the first and second phases, the ejector demo plant was operated with a reduced load (25% and 50%, respectively) and manual control; such a strategy was necessary to limit pressure and power consumption, since some demo plant devices showed lower performances than the one declared by the suppliers. Then, the demo plant entered the third and fourth phases of operation, in which the full load of the demo plant was reached. Nevertheless, in the same periods a growing issue related to mussels (*Mytilus galloprovincialis*) fouling in the pipes and filters has been detected. The performance of the demo plant was highly affected by fouling, since a reduced water flowrate was seen for the ejectors, and a higher pressure was needed, thus dramatically

increasing power consumption. This is why only 2 ejectors were in operation in the fifth phase.

| Demo plant operation regime | 2019 | | | | | | | 2020 | | | | | | | | |
|---|---|---|---|---|---|---|---|---|---|---|---|---|---|---|---|---|
| | Jun | Jul | Aug | Sep | Oct | Nov | Dec | Jan | Feb | Mar | Apr | May | Jun | Jul | Aug | Sep |
| Phase 1 - Manual, partial load (25% of maximum) | | | | | | | | | | | | | | | | |
| Phase 2 - Manual, partial load (50% of maximum) | | | | | | | | | | | | | | | | |
| Phase 3 - Manual, full load | | | | | | | | | | | | | | | | |
| Phase 4 - Automatic -10 ejectors | | | | | | | | | | | | | | | | |
| Phase 5 - Automatic - 2 ejectors | | | | | | | | | | | | | | | | |

**Figure 6.** Classification by phases of the ejector demo plant operation in Cervia.

Therefore, in order to assess the effectiveness of the ejector demo plant, only the bathymetric survey collected in the period from June 2019 (before ejector demo plant operation) to April 2020 have been analysed. The bathymetries realized after May 2020 refer to the demo plant in operation with only two ejectors and are not comparable with the previous ones.

For the whole operating period, energy consumption and ordinary and extraordinary maintenance activities have been measured and computed to also evaluate the technical and economic efficiency of the ejector demo plant. Furthermore, the environmental impact of the ejector demo plant has been assessed; in particular, the impacts on (i) integrity of seabed sediments and communities, (ii) underwater noise, and (iii) greenhouse gases (GHGs) and pollutant emissions have been evaluated. The results of these monitoring activities will be included in following papers.

### 3.2. Analysis of Bathymetries: Water Depth and Sediment Volume Variation over Time

The analysis was carried out over 3 years, starting from June 2017 and ending in June 2020, in order to investigate, for similar loadings in terms of wave climate and seasons, the sediment transport at the port entrance (i) with propeller operation and dredging (2017–2019) and (ii) during the operation of ejectors (2019–2020). The chosen periods are characterized by similar wave climates as shown in the frequency tables in Appendix A. All the bathymetries collected have been commissioned by the Municipality of Cervia and have been carried out through a digital hydrographic ultrasound system (Hydrotrac model, manufactured by Odom Hydrographic Systems, Baton Rouge, Louisiana, U.S.) with narrow emission cone, with the resulting error estimated as not exceeding 3 cm. The water depths reference is the mean water level.

Table 1 shows the 10 bathymetries (see Appendix B) considered for the aim of the paper, including timeframe and the relationship with sediment movimentation—i.e., dredging, propellers, and ejector demo plant operations.

**Table 1.** Bathymetries considered in the analysis for the period June 2017–April 2020.

| Bathymetry ID | Date | Note |
|---|---|---|
| 01 | 13 June 2017 | Realized after dredging (2200 m$^3$ of removed sediments) and 5 days of propellers operation completed in April–June 2017 |
| 02 | 28 December 2017 | |
| 03 | 7 April 2018 | |
| 04 | 11 May 2018 | Realized after 5 days of propeller operation, completed in April 2018 |
| 05 | 10 October 2018 | |
| 06 | 10 April 2019 | Realized after 2.5 days of propeller operation and dredging (20,000 m$^3$ of removed sediments), completed in January–April 2019 |
| 07 | 12 June 2019 | Realized one day before the ejector demo plant was put into in operation |
| 08 | 6 September 2019 | Ejector demo plant in operation (phase 1, see Figure 6) |
| 09 | 9 January 2020 | Ejector demo plant in operation (phase 2, see Figure 6) |
| 10 | 30 April 2020 | Ejector demo plant in operation (phase 3, see Figure 6) |

The bathymetries provided by the Municipality of Cervia include water depth measurements and the related coordination in AutoCAD files. QGIS 3.14 built-in Python was used by the authors to generate a model, while TIN interpolation, which is more useful for elevation, was used for interpolation; the assumed cell size was 5 m, the output raster Pixel sizes in X and Y are 0.1. The X and Y coordinate numbers are in the Project Coordinate Reference System (CRS) of WGS 84 (EPSG:4326) and the chosen unit is meters. In all the bathymetries, a common area can be identified (Figure 7).

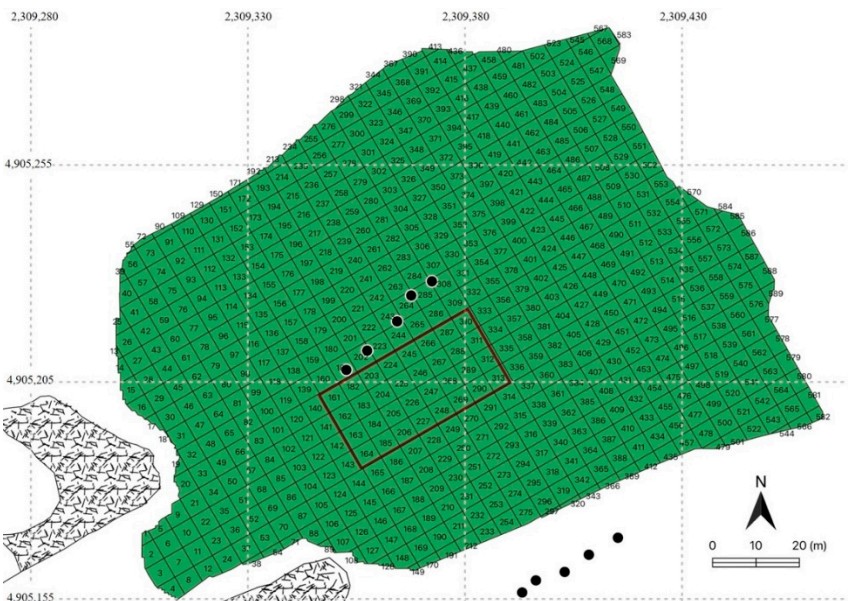

**Figure 7.** Grids in the common area of the analysed bathymetries. The box identifies the area primarily impacted by ejector operation. The black dots represent the mooring points of inlet (north) and outlet (south) ejectors' pipes.

The water depth measurements in this area are present in each analysed bathymetry. The definition of cells was necessary because comparing water depth variation at each measured point of the available bathymetries is rather impossible due to the relatively low accuracy in measured point replicability. By considering only the complete $5 \times 5$ m cells, that are 486, a whole area of 12,150 m$^2$ can be defined. The size of the cell ($5 \times 5$ m) is compatible with the area of the influence of one ejector. The red area identified in Figure 7 shows the area directly impacted by ejector demo plant operation, which is composed of 32 cells and measures 800 m$^2$. The black dots represent the mooring points of the inlet (north) and outlet (south, outside the surveyed area) ejectors' pipes (see also Figure 4).

By using the QGIS built-in Python, the average water depth and volume were defined for each cell at each bathymetry. The base level was assumed to be $-7$ m only for computing purposes. Then, water depth and volume variation can be compared over time by considering subsequent bathymetries. Any time dredging and/or propeller operation were performed, the relative bathymetry (i.e., bathymetry ID numbers 01, 04, and 06 in Table 1) was considered as the baseline for the following ones.

### 3.3. Analysis of 3-Year Metocean Climate on the Cervia Site

Since the annual net longshore sediment transport in the area offshore of the port of Cervia is estimated to be equal to zero [24], sea storms are the most significant driving forces leading to sediment transport and coastal changes; therefore, the identification of each single sea storm is necessary to assess the port sediment management in Cervia. Following the same analysis of the Nausicaa wave buoy performed in [29,30] (in Appendix C the time series of the wave height is reported), a sea storm is defined as an event characterized by a significant wave height higher than 1.5 m (i.e., the chosen threshold value) and remains

over this value for at least 6 h. Two storms are then considered as separate if the wave height decays below the threshold for 3 or more consecutive hours.

The study of marine weather events was completed with the calculation of the total energy E of each storm, identified through the integration of the significant height of the wave $H_s$ for the duration of the storm (*dur*), following the methodology performed in [31], subsequently adopted by [32] for local studies, to adapt the scale of ocean storms proposed by [30] to the Mediterranean context, as:

$$E = \int_{dur} H_s^2 dt \tag{1}$$

The event was then classified, following [31], through the energy classification scale defined in Table 2. The scope of the energy classification is to only compare periods in which the sea storm characterization is similar, based on the following assumptions: (i) the longshore sediment transport can be considered as rather constant in the area and (ii) the higher the storm energy registered in a certain period, the lower the contribution to sedimentation or erosion of longshore sediment transport.

**Table 2.** Energy-based classification of the sea storm.

| Class | Storm Energy (m$^2$ h) |
|---|---|
| I—weak | E ≤ 58.4 |
| II—moderate | 58.4 < E ≤ 127.9 |
| III—significant | 127.9 < E ≤ 389.7 |
| IV—severe | 389.7 < E ≤ 706.9 |
| V—extreme | E > 706.9 |

## 4. Results

### 4.1. Results of Meteoclimate Analysis

Table 3 reports a list and characteristics of the identified sea storms occurring in the period 2017–2020, together with the contemporary sea level and the maximum sea level during storm measured at a close tidal station, and with the bathymetry surveys and sediment movimentation actions at the port entrance.

The analysis was performed with the aim (i) of characterizing the extreme storm events mainly responsible for the short-term sediment movimentation occurring at the port entrance and (ii) of linking the amount of energy required for these events to the sediment movement at Cervia, in order to investigate the effectiveness of the ejector plant in 2019–2020, and to compare with the observations of the previous years (2017–2019), where no ejectors operated.

**Table 3.** Sea storm events in the period 2017–2020, together with the list of bathymetry surveys and sediment movimentation at the port entrance: peak wave significant height ($H_s$), mean ($T_m$), and peak ($T_p$) wave period, mean wave direction (MWD), compass sector, storm duration (*dur*), storm energy (E), energetic class, sea level at the $H_s$ instant, and maximum sea level during storm. [a] Sea level data not available; [b] Nausicaa data not available.

| Date | $H_s$ (m) | $T_m$ (s) | $T_p$ (s) | MWD (° N) | Compass Sector | *dur* (h) | E (m$^2$ h) | Class | Level (m) | Level Max (m) |
|---|---|---|---|---|---|---|---|---|---|---|
| April–June 2017 | | | | | Propeller movimentation | | | | | |
| 13 June 2017 | | | | | Bathymetry 01 | | | | | |
| 6 November 2017 | 2.79 | 5.6 | 8.3 | 61 | I | 19.5 | 89.68 | II—moderate | 0.53 | 0.83 |
| 13 November 2017 | 3.68 | 6.7 | 9.1 | 59 | I | 50.5 | 302.96 | III—significant | 0.44 | 0.93 |
| 26 November 2017 | 3.07 | 5.0 | 7.7 | 46 | I | 11 | 50.77 | I—weak | 0.13 | 0.36 |
| 2 December 2017 | 2.39 | 5.3 | 7.7 | 58 | I | 22 | 86.48 | II—moderate | 0.27 | 0.69 |

**Table 3.** *Cont.*

| Date | $H_s$ (m) | $T_m$ (s) | $T_p$ (s) | MWD (° N) | Compass Sector | *dur* (h) | E (m² h) | Class | Level (m) | Level Max (m) |
|---|---|---|---|---|---|---|---|---|---|---|
| 28 December 2017 | | | | | Bathymetry 02 | | | | | |
| 3 February 2018 | 2.51 | 5.3 | 8.3 | 55 | I | 9.5 | 36.15 | I—weak | 0.29 | 0.70 |
| 13 February 2018 | 1.78 | 4.4 | 6.2 | 24 | I | 7 | 20.33 | I—weak | 0.40 | 0.52 |
| 18 February 2018 | 2.70 | 5.6 | 8.3 | 59 | I | 15 | 70.10 | II—moderate | 0.10 | 0.45 |
| 24 February 2018 | 3.00 | 6.0 | 8.3 | 75 | I | 67.5 | 331.37 | III—significant | 0.36 | 0.70 |
| 26 February 2018 | 2.49 | 5.5 | 7.1 | 48 | I | 59 | 248.20 | III—significant | 0.28 | 0.61 |
| 21 March 2018 | 3.10 | 6.0 | 9.1 | 65 | I | 37 | 182.44 | III—significant | 0.43 | 0.83 |
| 23 March 2018 | 2.13 | 5.1 | 7.1 | 42 | I | 12.5 | 42.66 | I—weak | 0.38 | 0.72 |
| 7 April 2018 | | | | | Bathymetry 03 | | | | | |
| April 2018 | | | | | Propeller movimentation | | | | | |
| 11 May 2018 | | | | | Bathymetry 04 | | | | | |
| 26 August 2018 | 2.00 | 5.1 | 7.7 | 37 | I | 9 | 28.14 | I—weak | 0.27 | 0.53 |
| 24 September 2018 | 2.75 | 5.8 | 8.3 | 316 | IV | 47 | 188.87 | III—significant | 0.14 | 0.59 |
| 2 October 2018 | 2.36 | 5.3 | 7.7 | 23 | I | 11.5 | 49.20 | I—weak | 0.32 | 0.40 |
| 10 October 2018 | | | | | Bathymetry 05 | | | | | |
| 21 October 2018 | 2.76 | 5.6 | 7.1 | 340 | IV | 20 | 73.58 | II—moderate | 0.26 | 0.57 |
| 29 October 2018 | 2.63 | 6.2 | 9.1 | 46 | I | 16.5 | 75.98 | II—moderate | 0.79 | 1.06 |
| 17 November 2018 | 2.33 | 5.5 | 7.7 | 44 | I | 34.5 | 121.52 | II—moderate | [a] | [a] |
| 20 November 2018 | 2.66 | 5.4 | 7.7 | 42 | I | 11.5 | 55.93 | I—weak | [a] | [a] |
| 27 November 2018 | 2.30 | 5.1 | 6.2 | 66 | I | 16.5 | 53.12 | I—weak | [a] | [a] |
| January 2019 | | | | | Propeller movimentation | | | | | |
| 23 February 2019 | 2.84 | 6.1 | 9.1 | 66 | I | 32 | 145.29 | III—significant | −0.39 | 0.04 |
| 20 March 2019 | 1.89 | 4.8 | 7.1 | 63 | I | 6.5 | 20.10 | I—weak | −0.07 | 0.15 |
| 26 March 2019 | 3.60 | 6.3 | 8.3 | 38 | I | 7.5 | 67.50 | II—moderate | 0.56 | 0.83 |
| 4 April 2019 | 1.94 | 6.2 | 9.1 | 82 | I | 8 | 22.13 | I—weak | 0.34 | 0.57 |
| March–April 2019 | | | | | Dredging sand volume of 20,000 m³ | | | | | |
| 10 April 2019 | | | | | Bathymetry 06 | | | | | |
| 5 May 2019 | 2.77 | 5.6 | 7.1 | 52 | I | 19.5 | 80.04 | II—moderate | 0.42 | 0.48 |
| 12 May 2019 | 2.75 | 5.3 | 7.1 | 32 | I | 31 | 143.70 | III—significant | 0.30 | 0.46 |
| 14 May 2019 | 2.02 | 4.9 | 6.7 | 38 | I | 23.5 | 28.65 | I—weak | 0.07 | 0.14 |
| 12 January 2019 | | | | | Bathymetry 07 | | | | | |
| 18 January 2019 | | | | | Ejectors on | | | | | |
| 3 September 2019 | 1.85 | 4.8 | 6.7 | 55 | I | 6.5 | 18.8 | I—weak | 0.22 | 0.33 |
| 6 September 2019 | | | | | Bathymetry 08 | | | | | |
| 3 October 2019 | 2.5 | 5.4 | 7.7 | 28 | I | 9 | 38.4 | I—weak | 0.56 | 0.56 |
| 17 November 2019 | 1.87 | 6.5 | 9.1 | 83 | I | 7.5 | 22.8 | I—weak | 0.76 | 0.90 |
| 24 November 2019 | 1.77 | 5.5 | 8.3 | 82 | I | 13.5 | 34.8 | I—weak | 0.55 | 0.68 |
| 10 December 2019 | 1.75 | 5.1 | 6.7 | 66 | I | 11 | 30.9 | I—weak | 0.20 | 0.20 |
| 23 December 2019 | [b] | [b] | [b] | [b] | [b] | [b] | [b] | [b] | [b] | 1.0 |
| 9 January 2020 | | | | | Bathymetry 09 | | | | | |
| 20 January 2020 | 1.73 | 4.7 | 5.9 | 61 | I | 21 | 55.1 | I—weak | 0.12 | 0.13 |
| 6 February 2020 | 2.54 | 5.3 | 7.1 | 44 | I | 18.5 | 84.1 | II—moderate | −0.05 | 0.19 |
| 25 March 2020 | 4.05 | 7.8 | 25 | 58 | I | 76.5 | 369.6 | III—significant | −0.13 | 0.25 |
| 31 March 2020 | 2.24 | 5.2 | 7.1 | 48 | I | 17 | 56.0 | I—weak | 0.16 | 0.18 |
| 1 April 2020 | 1.88 | 4.7 | 6.2 | 61 | I | 10.5 | 29.9 | I—weak | 0.05 | 0.07 |
| 14 April 2020 | 2.58 | 5.5 | 7.7 | 59 | I | 16 | 58.5 | I—weak | −0.04 | 0.0 |
| 30 April 2020 | | | | | Bathymetry 10 | | | | | |

The yearly analyses of the sea storm events (Figure 8) show similarity among the considered years, with an average of 13 sea storm events per year. However, the seasonal distribution of the cumulated storm energy in Figure 9 is quite different between the 3 analysed years, showing:

- the most energetic year (E > 1400 m$^2$ h) was during the period 2017–2018, with the higher amount of storm energy in winter (E = 700 m$^2$ h) and then autumn (E = 520 m$^2$ h);
- the following year, 2018–2019, presented the most energetic sea in autumn (E > 600 m$^2$ h);
- the year 2019–2020 was the less energetic one (E $\cong$ 800 m$^2$ h), with higher amounts of energy in spring (E = 500 m$^2$ h);
- in all the years, small energy values were observed in summer, as expected.

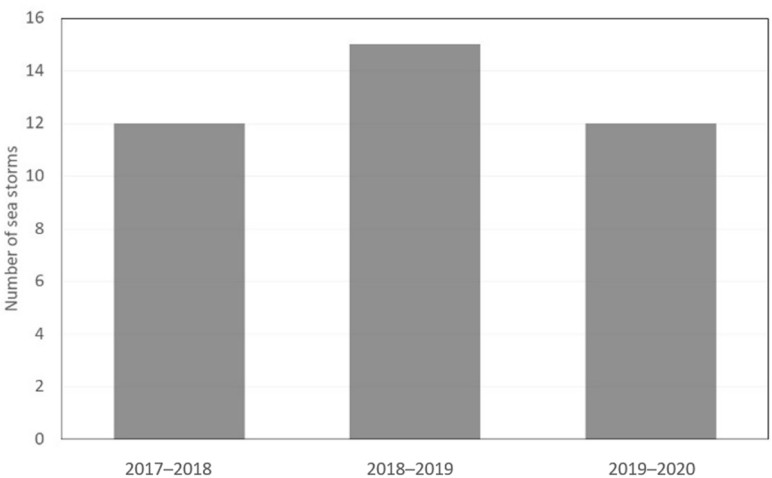

**Figure 8.** Annual number of sea storm events in the period 2017–2020.

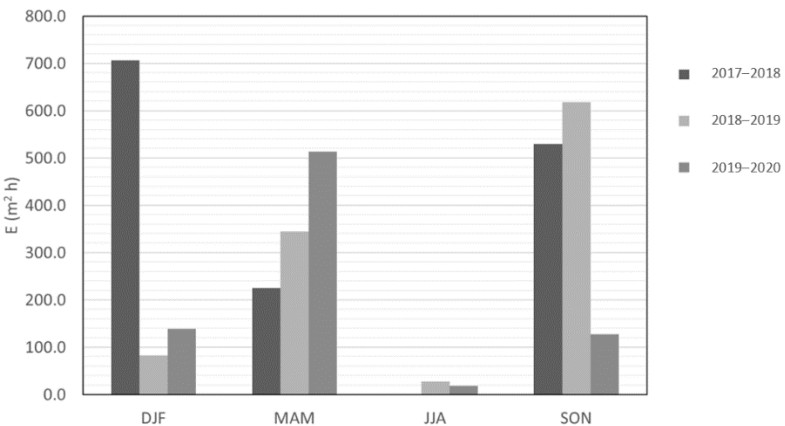

**Figure 9.** Seasonal distribution of the sea storm energy in the period 2017–2020: winter (DJF), spring (MAM), summer (JJA), and autumn (SON).

Although the overall wave conditions vary a lot over the years (see Figure 9), isolated periods with similar wave characteristics (in terms of cumulative storm energy) do occur, as highlighted in Table 4. For instance, the first (13 June 2017–28 December 2017) and last (9 January 2020–30 April 2020) periods analysed in Table 4 are characterized by a comparable number of sea storms (four and six, respectively) and amount of released energy (530 and 650 m$^2$ h, respectively).

**Table 4.** Summary of number of storms, related energy in periods under analysis, and the mean energy per day.

| Period | N° of Storms | Energy Released in the Period (m² h) | Mean Energy Per Day (m² h/day) |
|---|---|---|---|
| 13 June 2017–28 December 2017 | 4 | 530 | 2.68 |
| 28 December 2017–7 April 2018 | 7 | 930 | 9.30 |
| 11 May 2018–10 October 2018 | 3 | 266 | 1.75 |
| 10 April 2019–12 June 2019 | 4 | 275 | 4.37 |
| 12 June 2019–6 September 2019 | 1 | 19 | 0.22 |
| 6 September 2019–9 January 2020 | 4 | 127 | 1.02 |
| 9 January 2020–30 April 2020 | 6 | 653 | 5.83 |

### 4.2. Analysis of Water Depth Variation before and after Ejector Demo Plant Operation

After operation of propellers began in June 2017, the water depth at the port entrance appeared as reported in Figure 10, where the navigation channel presented a draft around −4 m, with 100 m width and 200 m length, while the northern area had a water depth greater than −2 m. The positions of the mooring points of the ejectors' inlet and outlet pipelines are also reported with black dots in Figure 10.

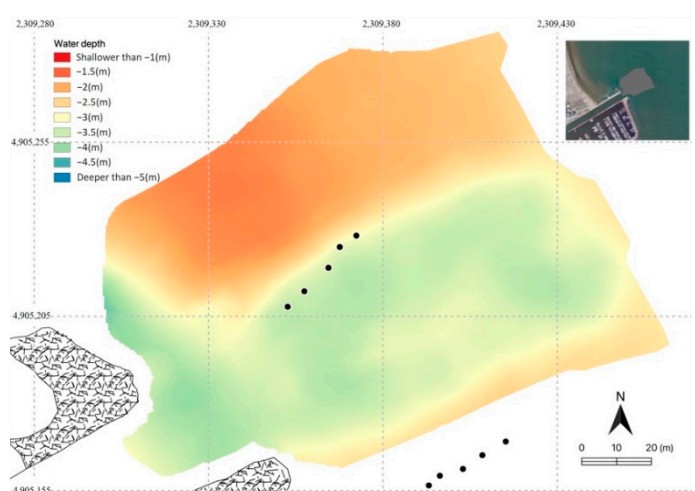

**Figure 10.** Maps of water depth at the port entrance after dredging and propeller operation on 13 June 2017. The positions of the mooring points of the ejectors' inlet and outlet pipelines are also reported as black dots.

Figure 11 shows the maps of water depth changes between two consecutive surveys in 2017–2018, where a hot–cold colour scale is representative of accumulated or eroded sediment volumes. In the first map, Figure 11a, seabed modification after summer and autumn seasons was observed, revealing in the first phase a sediment dynamic that mainly impacted the navigation channel, while in the second phase, the water depth variation was focused very close to the port docks.

In May 2018, a new propeller operation was needed since the port entrance was substantially closed—i.e., water depth under 2 m at the port inlet. Figure 12 shows the bathymetry realized after propeller operation, which is very similar to the one obtained in Figure 10. The water depth changes measured in the following bathymetry are shown in Figure 13 and reveal the same behaviour as observed in Figure 12—the sedimentation phenomenon is higher in the central navigation channel, while sediment moved from the surrounding areas.

From January to April 2019, the last dredging and propeller operation were realized before ejector demo plant installation. Figure 14 shows the bathymetry realized on 10 April of 2019. During this time, the sediment management activities affected a wider area,

including the entrance to the docks and the inner channel. Nevertheless, after 2 months, new sedimentation occurred in the area in front of the entrance to the docks, as already observed in the previous years (see Figure 15a).

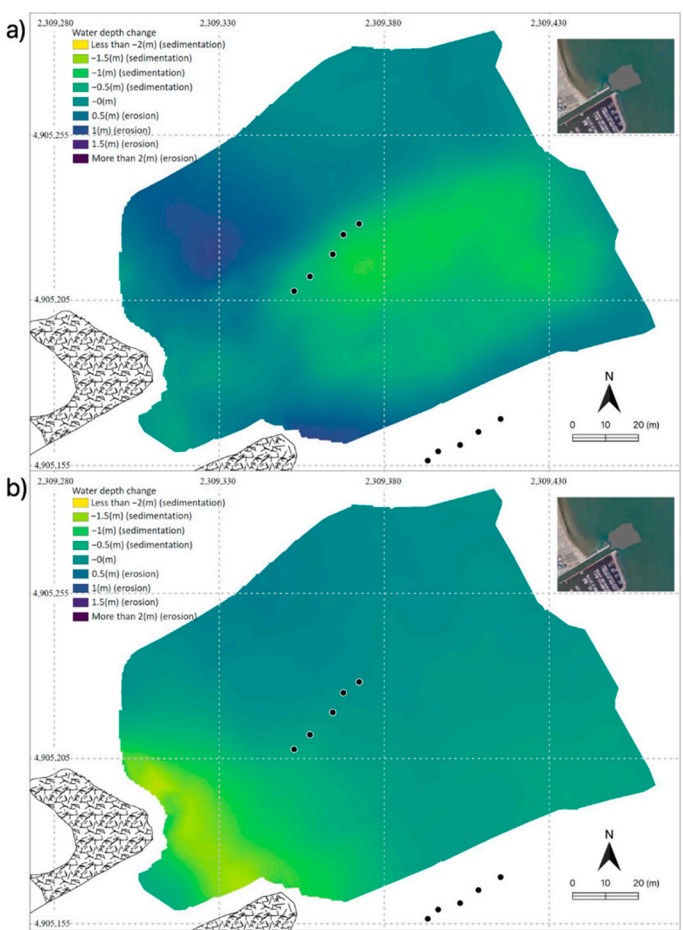

**Figure 11.** Water depth changes in the period: (**a**) 13 June 2017–28 December 2017 and (**b**) 28 December 2017–7 April 2018. The positions of the mooring points of the ejectors' inlet and outlet pipelines are also reported as black dots for reference purposes.

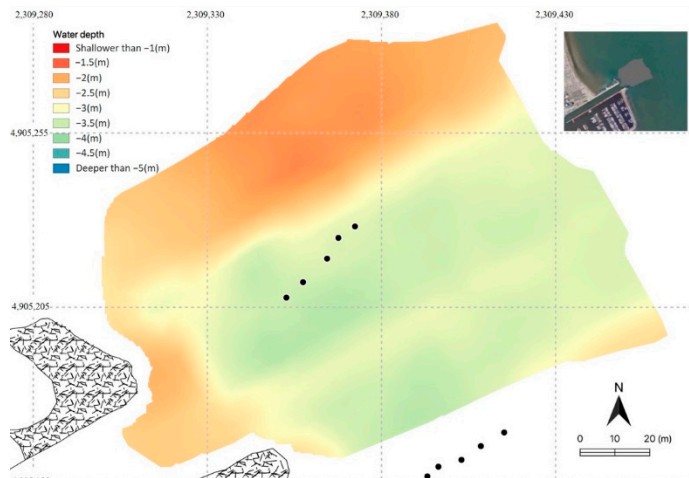

**Figure 12.** Maps of water depth at the port entrance after propeller operation on 11 May 2018. The positions of the mooring points of the ejectors' inlet and outlet pipelines are also reported as black dots.

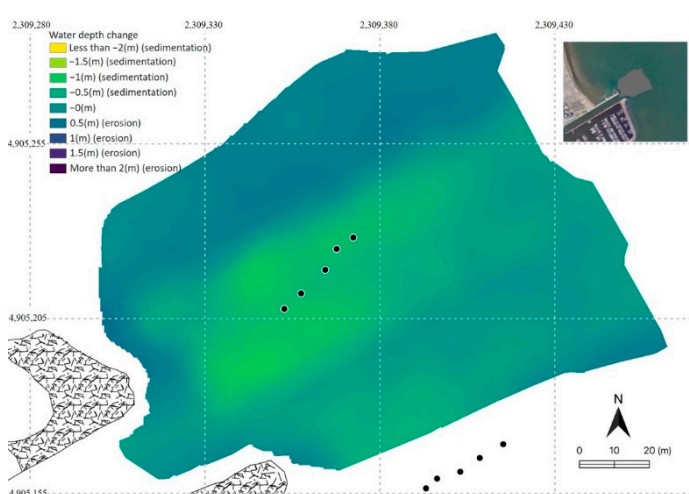

**Figure 13.** Water depth change in the period 11 May 2018–10 October 2018. The positions of the mooring points of the ejectors' inlet and outlet pipelines are also reported in black dots.

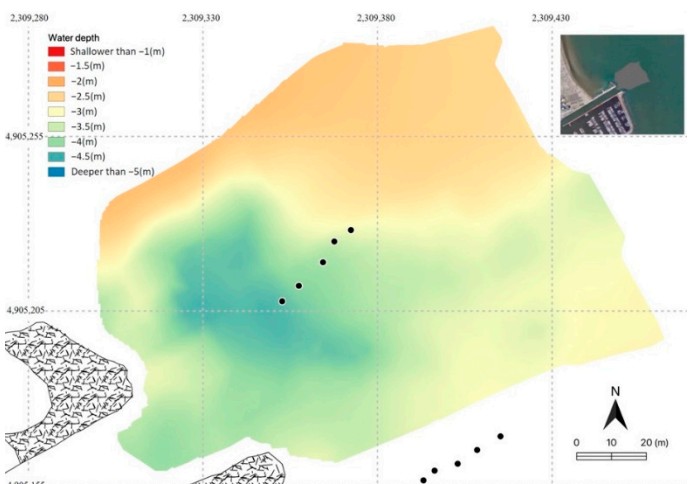

**Figure 14.** Maps of water depth at the port entrance after dredging and propeller operation on 10 April 2019. The positions of the mooring points of the ejectors' inlet and outlet pipelines are also reported as black dots.

On 13 June 2019 the ejector demo plant was activated. The bathymetry realized on 12 June 2019 (Figure 15b) is the reference bathymetry to evaluate ejector demo plant effectiveness in keeping a sufficient water level at the port entrance. The minimum water depth at the port entrance to guarantee navigability was 2.5 m, since under this level fisherman and leisure boats that use the Marina of Cervia usually start having navigability issues. Figure 16 has been realized on the basis of bathymetries from June 2019 to April 2020 (included in Appendix B) and shows if and where the minimum water depth was reached in the common area at the port inlet.

The first relevant result is that at the end of the monitoring period (end of April 2020) there is still present a navigable channel (i.e., water depth over 2.5 m) to enter the port of Cervia. It is interesting to note how in January 2020 the situation appeared critical in the area of influence of the ejectors, and that this critical situation negatively evolved from September 2019. Nevertheless, it should be considered that until February 2020, the ejector demo plant was not able to operate at full load, while starting from February 2020, the technical issues that limited the operation of the demo plant were solved.

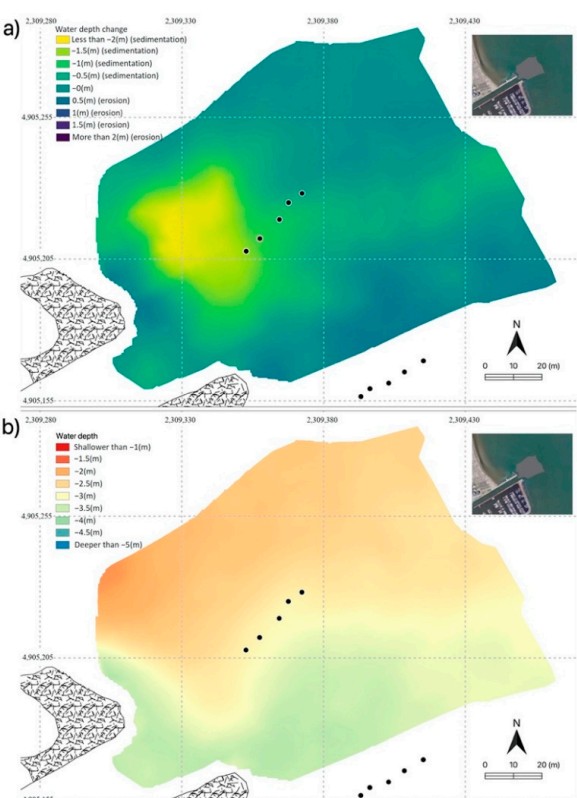

**Figure 15.** (**a**) Water depth change in the period 10 April 2019–12 June 2019, and (**b**) water depth measured on 12 June 2019. The positions of the mooring points of the ejectors' inlet and outlet pipelines are also reported as black dots.

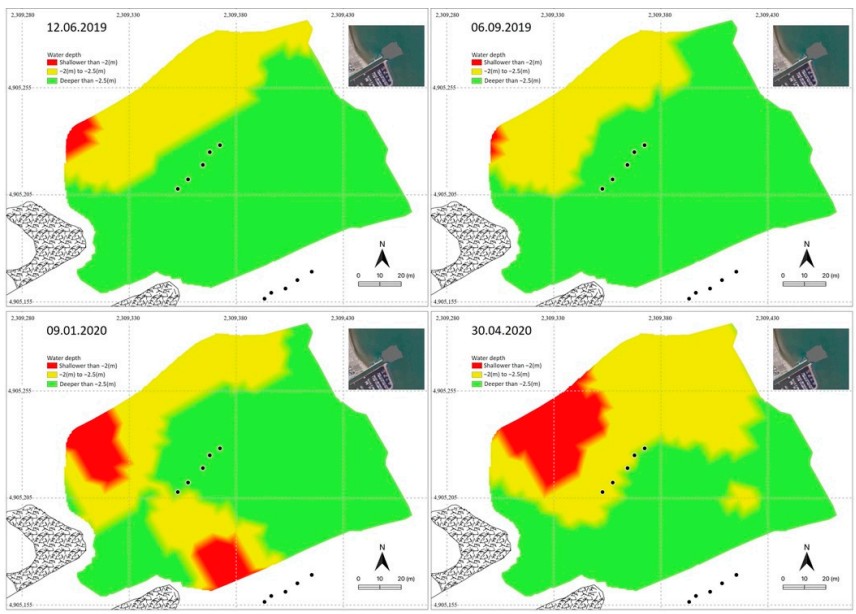

**Figure 16.** Critical areas (i.e., water depth < 2.5 m) during the ejector operation (12 June 2019–30 April 2020). The positions of the mooring points of the ejectors' inlet and outlet pipelines are also reported as black dots.

### 4.3. Analysis of Volume Variation before and after Ejector Demo Plant Operation

Volume variation over time has been considered in both areas shown in Figure 7 (common area and ejector area). The parameter computed for the comparison is the water

depth variation per day, expressed in mm per day, which is calculated by dividing the volume variation between two consecutive bathymetries by the area under consideration (i.e., common area or ejector area) and by the number of days between two consecutive bathymetries. The results are shown in Figure 17.

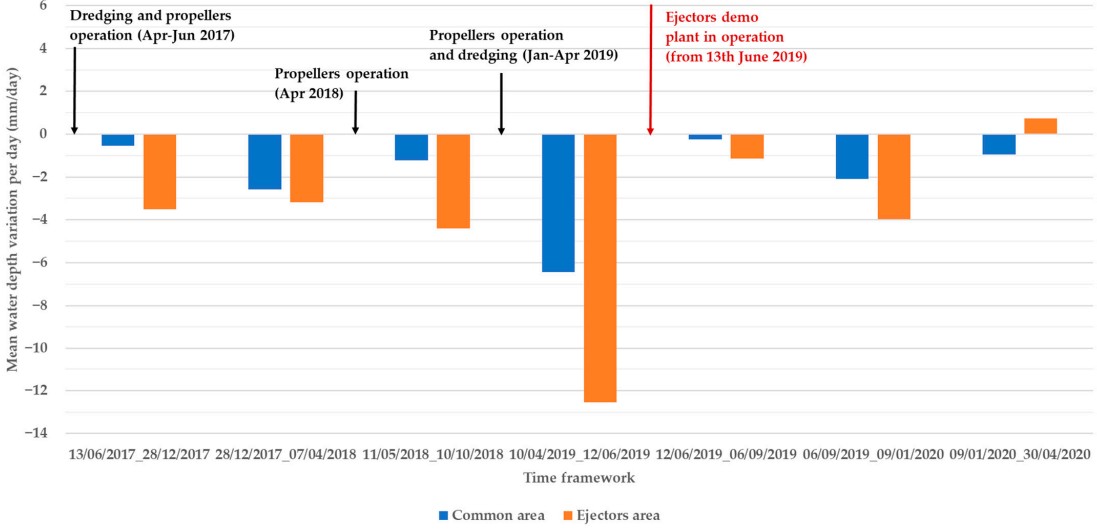

**Figure 17.** Mean water depth variation per day (in mm/day) between two consecutive bathymetries in the common area and in the ejector area before and after ejector demo plant operation.

The volume variation in both common and ejector areas has been related to the metocean data to evaluate if the positive water depth variation observed in Figure 17 can somehow be influenced by the natural sediment transport dynamic. In this case, the parameter computed for the comparison is the water depth variation per storm energy unit, expressed in mm per $m^2$ h, which is calculated by dividing the volume variation between two consecutive bathymetries by the area under consideration (i.e., common area or ejector area) and by the cumulated energy produced by the storms registered between two consecutive bathymetries (see Table 3 for metocean data). The results are shown in Figure 18.

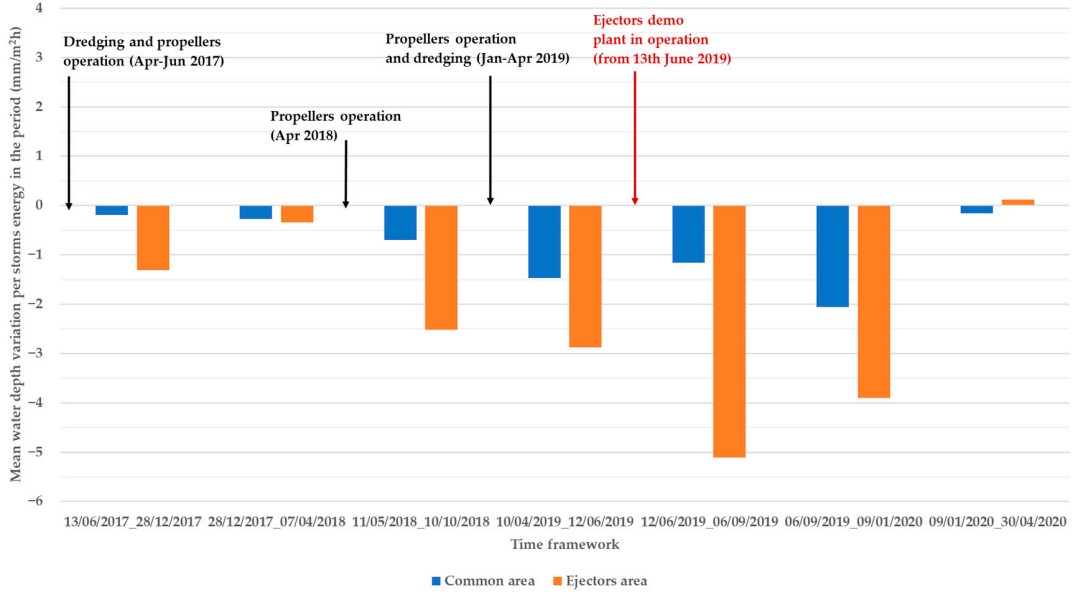

**Figure 18.** Mean water depth variation per energy of storms in the period (in mm/$m^2$ h) between two consecutive bathymetries in the common area and in the ejector area before and after ejector demo plant operation.

The implicit assumption in drafting Figure 18 is that sediment transport in both areas under investigation is mainly due to storms. Therefore, different periods considered in the analysis can be compared only if similar storm conditions occur.

## 5. Discussion

The analysis of Figures 10–15, it is clearly shown that dredging and propeller operation realized in the centre of the navigation channel partially solve the problem of navigability of the port inlet, since in a few months the hole created at the centre of the channel was covered again by sediment. In particular, part of the sediment came from the surrounding area, and it is expected that natural sediment transport in the area produced by storm and longshore transport also contributes. The impact of dredging and propeller operation on the water depth variation rate is also confirmed by the analysis of Figures 17 and 18. In particular, both figures show sediment dynamics that seem to impact more on the area of ejectors than in the whole common area. Such a dynamic is justified by the fact that the ejector area is directly affected by both dredging and propeller operation, meaning that after artificial deepening of the seabed, the ejector area is usually characterized by a water depth that is higher than the mean value of the common area. The result is that the ejector area works as a sediment trap after dredging or propeller operation. Moreover, in the ejector area, in the first period after dredging or propeller operation (13 June 2017–28 December 2017, 11 May 2018–10 October 2018, 10 April 2019–12 June 2019), we observed a higher water depth variation than in the common area, while in the period 28 December 2017–7 April 2018, which is characterized by the highest number of storms, related energy in the period as well as energy per day, the water depth variations in both areas are comparable. Furthermore, Figure 18 shows a relevant decreasing water depth variation rate per storm energy unit in the two consecutive periods from June to December 2017 and from December 2017 to April 2018, since in the first period the rate is $-1.31$ mm/m$^2$ h, while in the second period it is $-0.34$ mm/m$^2$ h. It can be concluded that after a faster variation of water depth in the ejector area, which is the one mainly affected by dredging and propeller operation, the water depth variation tends to homogenize in the common area. Therefore, the "artificial" increase in water depth modifies the natural sediment dynamic since the depression created in the port inlet attracts the nearby sediment and the sediment that is naturally transported in the area. A better option would be to not dredge or move the sediment via propeller operation along the navigable channel, but to work on the southern or northern areas. Such planning would be beneficial also in combination with ejector demo plant operation, since the dredging or propeller operation, which may be needed to remove sediment accumulation out of the ejector area for beach nourishment purposes, would not affect the integrity of the ejector demo plant.

While Figure 17 indicates that there is not a constant relationship between the water depth variation in the common and in the ejector areas, the same figure suggests that water depth variation in the common area before and after ejector demo plant operation has a comparable intensity, i.e., $-0.5 \div 2.5$ mm/day of mean variation, with the exception of the period 10 April 2019–12 June 2019, in which the water depth variation in the common area reaches $-6.5$ mm/day. Such a huge water depth variation rate is justified by the combined effects of (i) dredging and propeller operation realized on a wide area at the port entrance (see Figure 14) and of (ii) relative high storm energy measured in the period. The period 12 June 2019–6 September 2019 is characterized by a very low level of storm energy, which is almost zero, yielding the highest value in Figure 18. In this case, it is probable that the sediment transport has not only occurred due to the single storm registered in the period, but natural longshore sediment transport also contributes. Therefore, the longshore sediment dynamic is worth investigation in this area to better evaluate volume and direction of the sediment transport during nice weather periods.

The most interesting evidence of Figures 17 and 18 is that in the last period of operation of the ejector demo plant (9 January 2020–30 April 2020), which overlaps with phase 3 of operation (see Figure 6), a positive water depth variation (i.e., navigability increasing)

in the area of the ejectors can be observed, while in the common area, a negative water depth variation was observed in the same period. The last period of operation of the ejector demo plant is comparable to the first period 13 June 2017–28 December 2017, since the two periods were characterized by similar energetic forcing from sea and similar metocean characteristics (see Tables 3 and 4). It is interesting to note that, in the comparing period, there is a relevant negative water depth variation, especially if compared with the common area. This fact suggests that in this period the impact of ejector demo plant operation is evident and contributed to keep the water depth almost constant in the ejector area.

By assuming the same mean rate of water depth variation for the period 13 June 2017–28 December 2017, i.e., $-1.31$ mm/m$^2$ h, it can be estimated that, without the ejector demo plant in operation in the period 9 January 2020–30 April 2020, the water depth would vary by about $-0.855$ m in the ejector area, while a mean water depth variation of about 0.081 m has been observed. Therefore, the net contribution of the ejector demo plant can be evaluated with a maximum water depth variation of 0.936 m, which corresponds to a maximum volume of sediment by-passed of about 750 m$^3$.

Nevertheless, since the water depth variation rate in the period 13 June 2017–28 December 2017 may be influenced by the bathymetric changes produced in the area by the previous dredging operation, the potential impact of the ejectors has been evaluated by assuming the same mean rate of water depth variation of the following period, 28 December 2017–7 April 2018—i.e., $-0.34$ mm/m$^2$ h. If this water depth variation rate is applied, it can be estimated that without the ejector demo plant in operation in the period 9 January 2020–30 April 2020, the water depth would vary by about $-0.223$ m in the ejector area. In this case, the net contribution of the ejector demo plant can be evaluated in a maximum water depth variation of 0.304 m, which corresponds to a maximum volume of sediment by-passed of about 245 m$^3$.

While the positive impact in the ejector area produced by the ejector demo plant operation is evident in the period 9 January 2020–30 April 2020 and can be estimated, the same cannot be said for the previous periods. The different impacts are justified by the different operation regimes imposed on the ejector demo plant. In particular, the previous periods are characterized by lower water flowrates feeding the ejectors. Nevertheless, it is not possible to exclude that some contributions were made by the ejector demo plant operation in the periods of 12 June 2019–6 September 2019 and 6 September 2019–9 January 2020. Further investigation is needed to design a model of sediment dynamic in the port entrance, which will be validated by the bathymetries realized before demo plant operation, and then the sediment dynamic in the same metocean frameworks registered during the ejector demo plant operation will be simulated to verify the net contribution of the ejector demo plant to water depth variation.

## 6. Conclusions

An innovative technology for sediment management in water infrastructure has been tested in the first industrial sized demo plant at the port entrance of the Marina of Cervia (Italy). The monitoring activities, concluded in September 2020, involved several activities, which include effectiveness, efficacy, and environmental impact assessments. This paper investigates the effectiveness achieved and the results demonstrate that the ejector demo plant was able to guarantee navigability at the port inlet after almost one year of operation (June 2019–April 2020). In particular, the maximum impact of the ejector demo plant on keeping the water depth at the desired level (i.e., over the minimum threshold of 2.5 m) was observed in the period January–April 2020, wherein the ejector demo plant was able to operate at the design water feeding flowrates and with an estimated by-passed sediment volume of between 245 and 750 m$^3$.

Further investigation is needed to confirm the result through (i) the design of a model of sediment dynamics at the port entrance and (ii) the simulation of the sediment transport in the same metocean frameworks registered during the ejector demo plant operation in order to confirm the contribution on water depth and sediment volume variation. The

activities will be carried out once the monitoring of water depth at the port entrance, which is still ongoing, in the second half of 2021.

Finally, based on the good performance of the ejector demo plant shown in the last period of operation (January–April 2020), an adaptation of the existing ejectors configuration is under evaluation to optimize demo plant operation. The hypothesis is to space out the ejectors and, at the same time, move them closer to the port entrance. Another hypothesis under investigation is reducing the number of ejectors installed to reduce whole power consumption.

**Author Contributions:** Conceptualization, M.P. and M.G.G.; methodology, A.A., M.P. and M.G.G.; software, A.A.; formal analysis, A.A. and M.G.G.; investigation, A.A. and M.G.G.; data curation, A.A.; writing—original draft preparation, M.G.G., A.A. and M.P.; writing—review and editing, R.A., A.G. and C.S.; visualization, A.A.; supervision, R.A. and C.S. All authors have read and agreed to the published version of the manuscript.

**Funding:** This research was funded by the Italian Ministry for the Environment and Protection of the Territory and the Sea, (MATTM) under the STIMARE Project, grant number CUP J56C18001240001.

**Institutional Review Board Statement:** Not applicable.

**Informed Consent Statement:** Not applicable.

**Data Availability Statement:** Data can be made available by the authors based on request.

**Acknowledgments:** The study is part of the "STIMARE (Strategie Innovative per il Monitoraggio ed Analisi del Rischio Erosione, www.progettostimare.it (accessed on 19 December 2020))" project, funded by the Italian Ministry for the Environment and Protection of the Territory and the Sea, which aims to study the shoreline evolution presence of coastal defence structures with innovative monitoring techniques and strategies.

**Conflicts of Interest:** The authors declare no conflict of interest.

## Appendix A. Frequency Tables of the Sea Climate

**Table A1.** Frequency table of significant wave heights $H_s$ and mean direction MWD for the year 2017–2018.

| MWD [°N] | Hs [m] | | | | | | | | | | | | |
|---|---|---|---|---|---|---|---|---|---|---|---|---|---|
| | <0.50 | 0.50–0.75 | 0.75–1.00 | 1.00–1.25 | 1.25–1.5 | 1.5–1.75 | 1.75–2.00 | 2.00–2.25 | 2.25–2.5 | 2.5–2.75 | 2.75–3 | >3.00 | Total |
| 0–15 | 2.43 | 0.36 | 0.10 | 0.03 | 0.01 | 0.00 | 0.01 | 0.00 | 0.00 | 0.00 | 0.00 | 0.00 | 2.94 |
| 15–30 | 3.27 | 0.78 | 0.26 | 0.24 | 0.04 | 0.09 | 0.06 | 0.00 | 0.03 | 0.00 | 0.00 | 0.00 | 4.78 |
| 30–45 | 4.36 | 1.28 | 0.54 | 0.53 | 0.34 | 0.22 | 0.27 | 0.10 | 0.03 | 0.02 | 0.01 | 0.01 | 7.71 |
| 45–60 | 3.79 | 1.32 | 1.05 | 0.79 | 0.63 | 0.40 | 0.49 | 0.58 | 0.48 | 0.15 | 0.02 | 0.01 | 9.70 |
| 60–75 | 4.61 | 1.33 | 0.81 | 0.52 | 0.39 | 0.29 | 0.43 | 0.63 | 0.25 | 0.21 | 0.21 | 0.02 | 9.70 |
| 75–90 | 9.64 | 2.44 | 1.36 | 1.25 | 0.87 | 0.16 | 0.01 | 0.01 | 0.00 | 0.00 | 0.00 | 0.01 | 15.74 |
| 90–105 | 16.46 | 3.73 | 1.37 | 0.46 | 0.18 | 0.07 | 0.00 | 0.00 | 0.00 | 0.00 | 0.00 | 0.00 | 22.26 |
| 105–120 | 10.53 | 1.42 | 0.26 | 0.11 | 0.00 | 0.00 | 0.00 | 0.00 | 0.00 | 0.00 | 0.00 | 0.00 | 12.32 |
| 120–135 | 2.23 | 0.16 | 0.06 | 0.00 | 0.00 | 0.00 | 0.00 | 0.00 | 0.00 | 0.00 | 0.00 | 0.00 | 2.45 |
| 135–150 | 0.54 | 0.03 | 0.00 | 0.00 | 0.00 | 0.00 | 0.00 | 0.00 | 0.00 | 0.00 | 0.00 | 0.00 | 0.58 |
| 150–165 | 0.20 | 0.00 | 0.00 | 0.00 | 0.00 | 0.00 | 0.00 | 0.00 | 0.00 | 0.00 | 0.00 | 0.00 | 0.20 |
| 165–180 | 0.07 | 0.01 | 0.00 | 0.00 | 0.00 | 0.00 | 0.00 | 0.00 | 0.00 | 0.00 | 0.00 | 0.00 | 0.08 |
| 180–195 | 0.15 | 0.01 | 0.00 | 0.00 | 0.00 | 0.00 | 0.00 | 0.00 | 0.00 | 0.00 | 0.00 | 0.00 | 0.16 |
| 195–210 | 0.17 | 0.00 | 0.00 | 0.00 | 0.00 | 0.00 | 0.00 | 0.00 | 0.00 | 0.00 | 0.00 | 0.00 | 0.17 |
| 210–225 | 0.11 | 0.00 | 0.00 | 0.00 | 0.00 | 0.00 | 0.00 | 0.00 | 0.00 | 0.00 | 0.00 | 0.00 | 0.11 |
| 225–240 | 0.10 | 0.00 | 0.00 | 0.00 | 0.00 | 0.00 | 0.00 | 0.00 | 0.00 | 0.00 | 0.00 | 0.00 | 0.10 |

**Table A1.** *Cont.*

| MWD [°N] | Hs [m] | | | | | | | | | | | | Total |
|---|---|---|---|---|---|---|---|---|---|---|---|---|---|
| | <0.50 | 0.50–0.75 | 0.75–1.00 | 1.00–1.25 | 1.25–1.5 | 1.5–1.75 | 1.75–2.00 | 2.00–2.25 | 2.25–2.5 | 2.5–2.75 | 2.75–3 | >3.00 | |
| 240–255 | 0.27 | 0.08 | 0.00 | 0.00 | 0.00 | 0.00 | 0.00 | 0.00 | 0.00 | 0.00 | 0.00 | 0.00 | 0.35 |
| 255–270 | 0.32 | 0.00 | 0.00 | 0.00 | 0.01 | 0.00 | 0.00 | 0.00 | 0.00 | 0.00 | 0.00 | 0.00 | 0.33 |
| 270–285 | 0.68 | 0.00 | 0.00 | 0.00 | 0.00 | 0.00 | 0.00 | 0.00 | 0.00 | 0.00 | 0.00 | 0.00 | 0.68 |
| 285–300 | 0.98 | 0.04 | 0.03 | 0.00 | 0.00 | 0.00 | 0.00 | 0.00 | 0.00 | 0.00 | 0.00 | 0.00 | 1.05 |
| 300–315 | 1.71 | 0.05 | 0.00 | 0.02 | 0.00 | 0.00 | 0.00 | 0.00 | 0.00 | 0.00 | 0.00 | 0.00 | 1.78 |
| 315–330 | 1.69 | 0.03 | 0.06 | 0.01 | 0.00 | 0.00 | 0.00 | 0.00 | 0.00 | 0.00 | 0.00 | 0.00 | 1.79 |
| 330–345 | 1.63 | 0.13 | 0.07 | 0.02 | 0.00 | 0.00 | 0.00 | 0.00 | 0.00 | 0.00 | 0.00 | 0.00 | 1.84 |
| 345–360 | 2.68 | 0.33 | 0.09 | 0.07 | 0.01 | 0.00 | 0.00 | 0.00 | 0.00 | 0.00 | 0.00 | 0.00 | 3.18 |
| Total | 68.62 | 13.53 | 6.06 | 4.05 | 2.48 | 1.23 | 1.26 | 1.32 | 0.79 | 0.37 | 0.24 | 0.05 | 100 |

**Table A2.** Frequency table of significant wave height $H_s$ and mean direction MWD for the year 2018–2019.

| MWD [°N] | Hs [m] | | | | | | | | | | | | Total |
|---|---|---|---|---|---|---|---|---|---|---|---|---|---|
| | <0.50 | 0.50–0.75 | 0.75–1.00 | 1.00–1.25 | 1.25–1.5 | 1.5–1.75 | 1.75–2.00 | 2.00–2.25 | 2.25–2.5 | 2.5–2.75 | 2.75–3 | >3.00 | |
| 0–15 | 3.16 | 0.53 | 0.16 | 0.06 | 0.03 | 0.02 | 0.01 | 0.05 | 0.03 | 0.01 | 0.01 | 0.00 | 4.06 |
| 15–30 | 4.78 | 1.13 | 0.55 | 0.20 | 0.12 | 0.02 | 0.02 | 0.03 | 0.05 | 0.02 | 0.01 | 0.00 | 6.92 |
| 30–45 | 4.86 | 1.38 | 0.99 | 0.57 | 0.37 | 0.26 | 0.21 | 0.19 | 0.06 | 0.04 | 0.01 | 0.00 | 8.94 |
| 45–60 | 5.34 | 1.49 | 1.39 | 0.92 | 0.79 | 0.39 | 0.32 | 0.15 | 0.07 | 0.04 | 0.01 | 0.00 | 10.92 |
| 60–75 | 5.42 | 1.34 | 1.35 | 1.08 | 0.95 | 0.50 | 0.23 | 0.11 | 0.08 | 0.09 | 0.02 | 0.00 | 11.16 |
| 75–90 | 6.02 | 1.31 | 0.89 | 0.86 | 0.43 | 0.22 | 0.03 | 0.02 | 0.00 | 0.00 | 0.00 | 0.00 | 9.80 |
| 90–105 | 9.81 | 2.56 | 1.01 | 0.49 | 0.24 | 0.05 | 0.01 | 0.00 | 0.00 | 0.00 | 0.00 | 0.00 | 14.19 |
| 105–120 | 5.89 | 0.86 | 0.31 | 0.04 | 0.01 | 0.06 | 0.03 | 0.00 | 0.00 | 0.00 | 0.00 | 0.00 | 7.21 |
| 120–135 | 3.07 | 0.25 | 0.05 | 0.00 | 0.01 | 0.01 | 0.02 | 0.00 | 0.00 | 0.00 | 0.00 | 0.00 | 3.40 |
| 135–150 | 2.21 | 0.10 | 0.02 | 0.00 | 0.01 | 0.00 | 0.00 | 0.00 | 0.00 | 0.00 | 0.00 | 0.00 | 2.33 |
| 150–165 | 1.54 | 0.06 | 0.01 | 0.00 | 0.01 | 0.01 | 0.01 | 0.01 | 0.00 | 0.00 | 0.00 | 0.00 | 1.65 |
| 165–180 | 1.41 | 0.02 | 0.01 | 0.00 | 0.01 | 0.01 | 0.01 | 0.00 | 0.00 | 0.00 | 0.00 | 0.00 | 1.46 |
| 180–195 | 1.09 | 0.03 | 0.00 | 0.00 | 0.00 | 0.00 | 0.01 | 0.00 | 0.00 | 0.00 | 0.00 | 0.00 | 1.13 |
| 195–210 | 0.73 | 0.01 | 0.01 | 0.00 | 0.00 | 0.00 | 0.01 | 0.00 | 0.00 | 0.00 | 0.00 | 0.00 | 0.75 |
| 210–225 | 0.95 | 0.01 | 0.01 | 0.00 | 0.00 | 0.00 | 0.00 | 0.00 | 0.00 | 0.00 | 0.00 | 0.00 | 0.96 |
| 225–240 | 0.84 | 0.00 | 0.01 | 0.00 | 0.01 | 0.00 | 0.00 | 0.01 | 0.00 | 0.00 | 0.00 | 0.00 | 0.86 |
| 240–255 | 0.70 | 0.01 | 0.00 | 0.00 | 0.00 | 0.00 | 0.00 | 0.00 | 0.00 | 0.00 | 0.00 | 0.00 | 0.71 |
| 255–270 | 0.95 | 0.00 | 0.01 | 0.00 | 0.00 | 0.00 | 0.00 | 0.00 | 0.00 | 0.01 | 0.00 | 0.00 | 0.96 |
| 270–285 | 1.13 | 0.01 | 0.00 | 0.00 | 0.00 | 0.00 | 0.01 | 0.00 | 0.00 | 0.00 | 0.00 | 0.00 | 1.14 |
| 285–300 | 1.64 | 0.02 | 0.01 | 0.00 | 0.00 | 0.01 | 0.00 | 0.00 | 0.00 | 0.01 | 0.01 | 0.00 | 1.69 |
| 300–315 | 1.93 | 0.06 | 0.00 | 0.00 | 0.01 | 0.02 | 0.01 | 0.02 | 0.01 | 0.01 | 0.00 | 0.00 | 2.06 |
| 315–330 | 1.81 | 0.08 | 0.02 | 0.00 | 0.01 | 0.02 | 0.03 | 0.05 | 0.01 | 0.00 | 0.01 | 0.00 | 2.03 |
| 330–345 | 2.00 | 0.08 | 0.01 | 0.00 | 0.03 | 0.01 | 0.04 | 0.03 | 0.01 | 0.00 | 0.01 | 0.00 | 2.21 |
| 345–360 | 2.95 | 0.16 | 0.10 | 0.05 | 0.03 | 0.05 | 0.05 | 0.05 | 0.02 | 0.01 | 0.00 | 0.00 | 3.47 |
| Total | 70.22 | 11.49 | 6.90 | 4.28 | 3.06 | 1.64 | 1.06 | 0.72 | 0.34 | 0.22 | 0.08 | 0.00 | 100 |

**Table A3.** Frequency table of significant wave height $H_s$ and mean direction MWD for the year 2019–2020.

| MWD [°N] | Hs [m] | | | | | | | | | | | | |
|---|---|---|---|---|---|---|---|---|---|---|---|---|---|
| | <0.50 | 0.50–0.75 | 0.75–1.00 | 1.00–1.25 | 1.25–1.5 | 1.5–1.75 | 1.75–2.00 | 2.00–2.25 | 2.25–2.5 | 2.5–2.75 | 2.75–3 | >3.00 | Total |
| 0–15 | 3.27 | 0.41 | 0.09 | 0.03 | 0.03 | 0.01 | 0.01 | 0.01 | 0.01 | 0.00 | 0.00 | 0.00 | 3.86 |
| 15–30 | 4.40 | 0.55 | 0.17 | 0.10 | 0.02 | 0.02 | 0.02 | 0.02 | 0.02 | 0.01 | 0.00 | 0.00 | 5.32 |
| 30–45 | 4.38 | 0.78 | 0.56 | 0.24 | 0.10 | 0.08 | 0.08 | 0.10 | 0.05 | 0.02 | 0.02 | 0.00 | 6.42 |
| 45–60 | 3.90 | 1.39 | 1.42 | 0.97 | 0.52 | 0.49 | 0.25 | 0.20 | 0.14 | 0.17 | 0.01 | 0.00 | 9.46 |
| 60–75 | 4.91 | 1.49 | 2.00 | 0.86 | 0.50 | 0.66 | 0.18 | 0.10 | 0.08 | 0.04 | 0.00 | 0.00 | 10.84 |
| 75–90 | 7.48 | 2.11 | 1.33 | 0.93 | 0.63 | 0.24 | 0.05 | 0.00 | 0.01 | 0.00 | 0.00 | 0.00 | 12.77 |
| 90–105 | 17.20 | 4.54 | 1.32 | 0.42 | 0.04 | 0.02 | 0.01 | 0.00 | 0.00 | 0.00 | 0.00 | 0.00 | 23.55 |
| 105–120 | 10.82 | 1.07 | 0.25 | 0.04 | 0.01 | 0.00 | 0.00 | 0.00 | 0.00 | 0.00 | 0.00 | 0.00 | 12.19 |
| 120–135 | 2.79 | 0.35 | 0.07 | 0.02 | 0.00 | 0.00 | 0.00 | 0.00 | 0.00 | 0.00 | 0.00 | 0.00 | 3.23 |
| 135–150 | 0.33 | 0.02 | 0.02 | 0.00 | 0.00 | 0.00 | 0.00 | 0.00 | 0.00 | 0.00 | 0.00 | 0.00 | 0.37 |
| 150–165 | 0.09 | 0.00 | 0.00 | 0.00 | 0.00 | 0.00 | 0.00 | 0.00 | 0.00 | 0.00 | 0.00 | 0.00 | 0.09 |
| 165–180 | 0.09 | 0.00 | 0.00 | 0.00 | 0.00 | 0.00 | 0.00 | 0.00 | 0.00 | 0.00 | 0.00 | 0.00 | 0.09 |
| 180–195 | 0.08 | 0.01 | 0.00 | 0.00 | 0.00 | 0.00 | 0.00 | 0.00 | 0.00 | 0.00 | 0.00 | 0.00 | 0.09 |
| 195–210 | 0.09 | 0.00 | 0.00 | 0.00 | 0.00 | 0.00 | 0.00 | 0.00 | 0.00 | 0.00 | 0.00 | 0.00 | 0.09 |
| 210–225 | 0.05 | 0.01 | 0.00 | 0.00 | 0.00 | 0.00 | 0.00 | 0.00 | 0.00 | 0.00 | 0.00 | 0.00 | 0.06 |
| 225–240 | 0.14 | 0.04 | 0.00 | 0.00 | 0.00 | 0.00 | 0.00 | 0.00 | 0.00 | 0.00 | 0.00 | 0.00 | 0.18 |
| 240–255 | 0.28 | 0.06 | 0.00 | 0.00 | 0.00 | 0.00 | 0.00 | 0.00 | 0.00 | 0.00 | 0.00 | 0.00 | 0.34 |
| 255–270 | 0.38 | 0.03 | 0.01 | 0.00 | 0.00 | 0.00 | 0.00 | 0.00 | 0.00 | 0.00 | 0.00 | 0.00 | 0.42 |
| 270–285 | 0.46 | 0.04 | 0.00 | 0.00 | 0.00 | 0.00 | 0.00 | 0.00 | 0.00 | 0.00 | 0.00 | 0.00 | 0.50 |
| 285–300 | 0.91 | 0.01 | 0.00 | 0.00 | 0.00 | 0.00 | 0.00 | 0.00 | 0.00 | 0.00 | 0.00 | 0.00 | 0.91 |
| 300–315 | 1.52 | 0.02 | 0.00 | 0.00 | 0.01 | 0.00 | 0.00 | 0.00 | 0.00 | 0.00 | 0.00 | 0.00 | 1.55 |
| 315–330 | 1.78 | 0.03 | 0.02 | 0.01 | 0.01 | 0.00 | 0.00 | 0.00 | 0.00 | 0.00 | 0.00 | 0.00 | 1.84 |
| 330–345 | 1.77 | 0.08 | 0.03 | 0.04 | 0.00 | 0.00 | 0.00 | 0.00 | 0.00 | 0.00 | 0.00 | 0.00 | 1.91 |
| 345–360 | 3.43 | 0.29 | 0.17 | 0.02 | 0.01 | 0.00 | 0.00 | 0.00 | 0.00 | 0.00 | 0.00 | 0.00 | 3.92 |
| Total | 70.57 | 13.31 | 7.47 | 3.67 | 1.88 | 1.50 | 0.60 | 0.43 | 0.31 | 0.23 | 0.03 | 0.00 | 100 |

## Appendix B. Bathymetric Maps of the Study Area

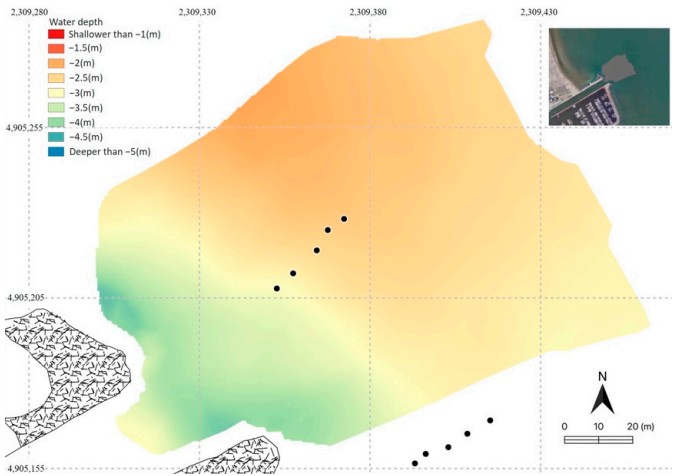

**Figure A1.** Bathymetry of 28 December 2017.

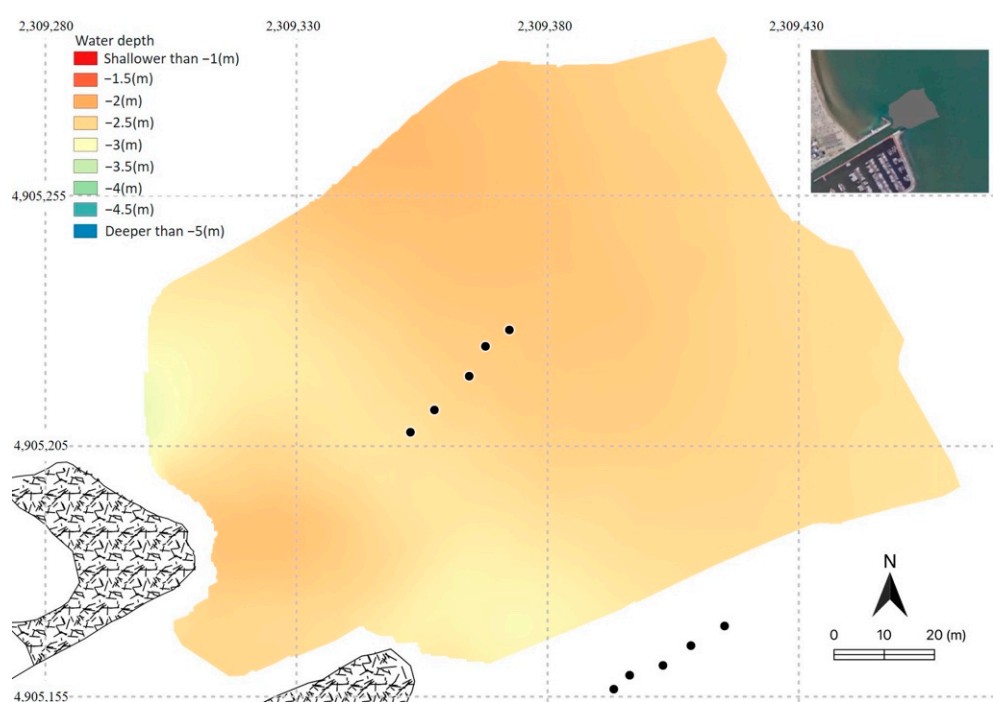

**Figure A2.** Bathymetry of 7 April 2018.

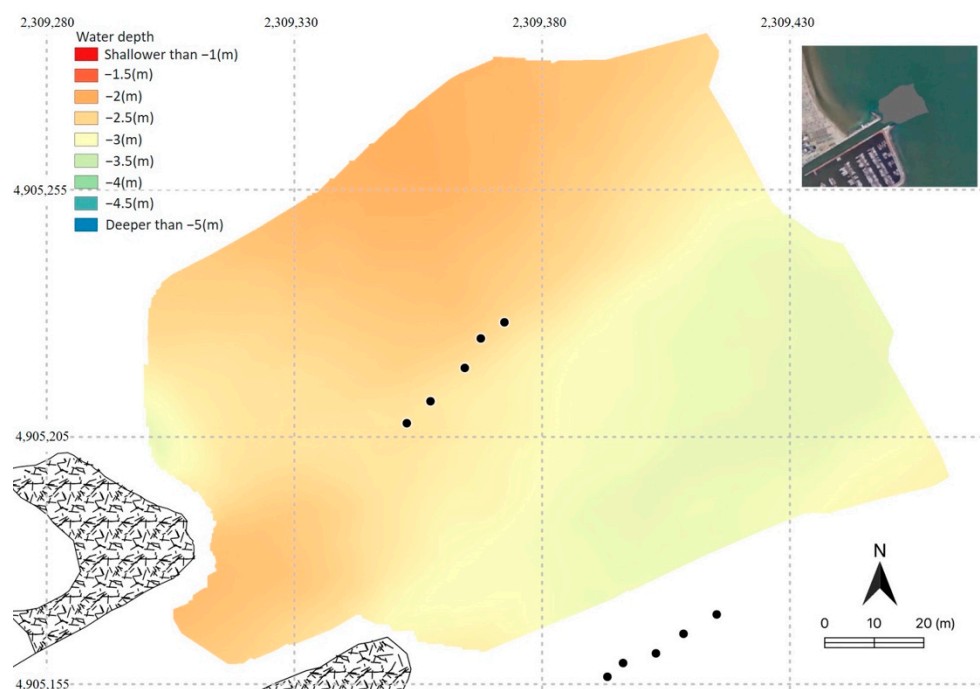

**Figure A3.** Bathymetry of 10 October 2018.

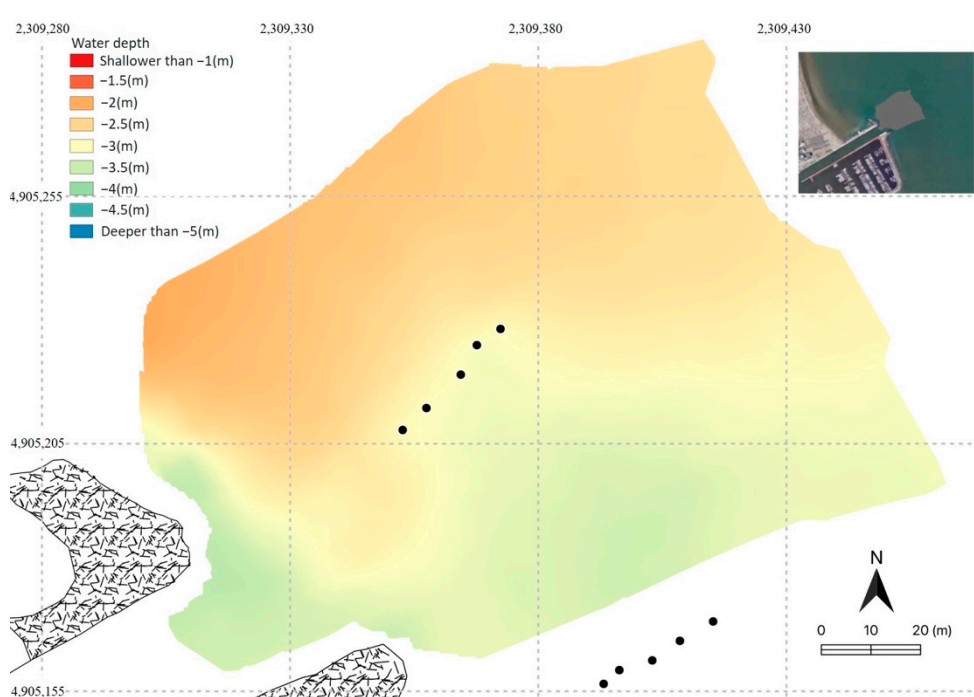

**Figure A4.** Bathymetry of 6 September 2019.

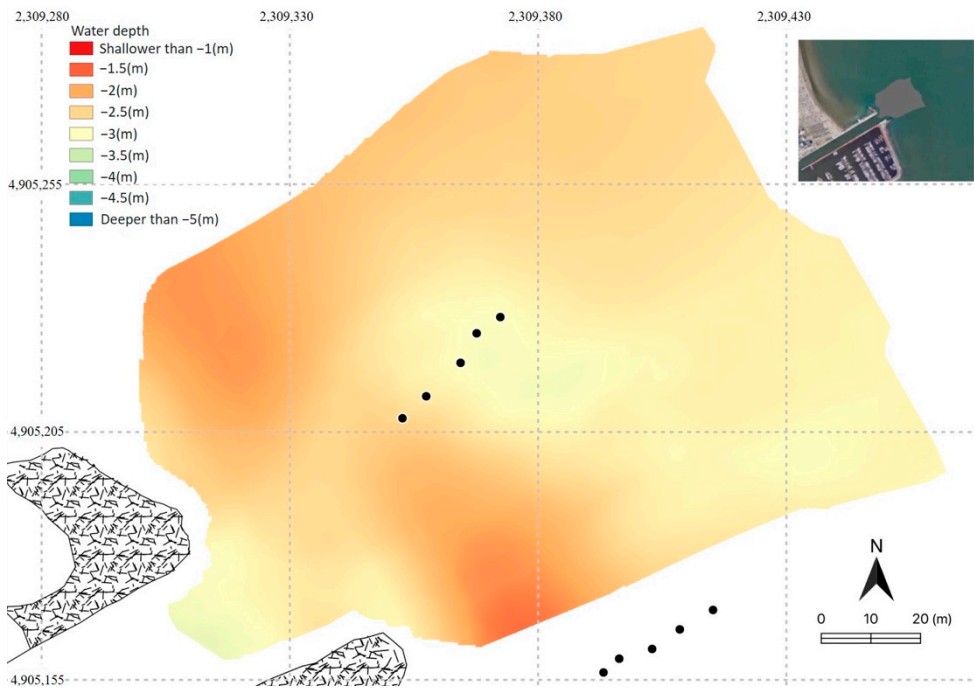

**Figure A5.** Bathymetry of 9 January 2020.

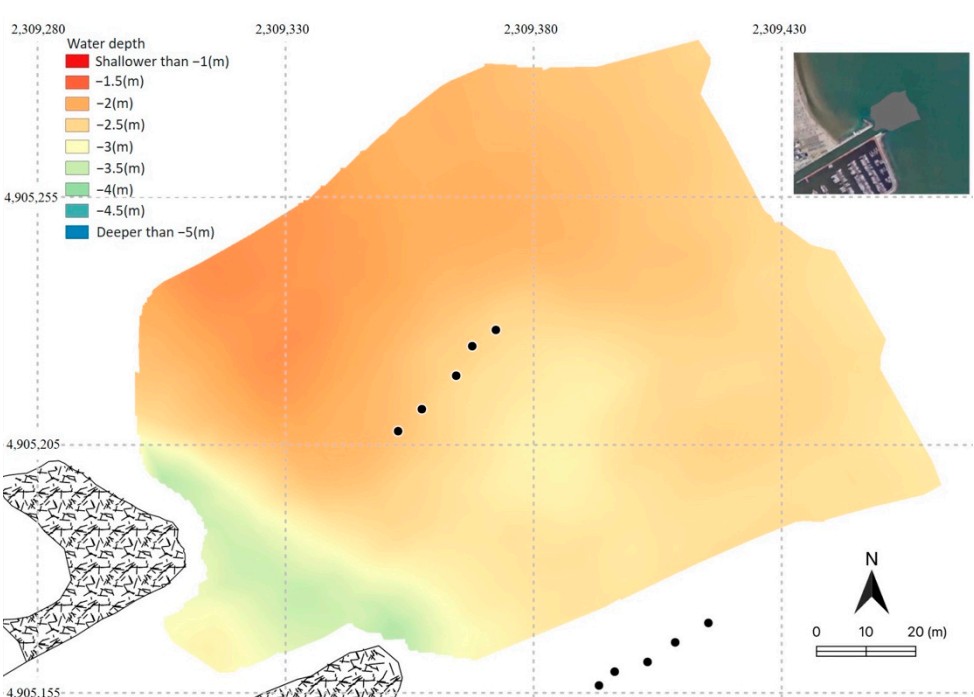

**Figure A6.** Bathymetry of 30 April 2020.

**Appendix C. Time Series of the Wave Height with the Timeline of the Realized Surveys and Sediment Movimentation Actions**

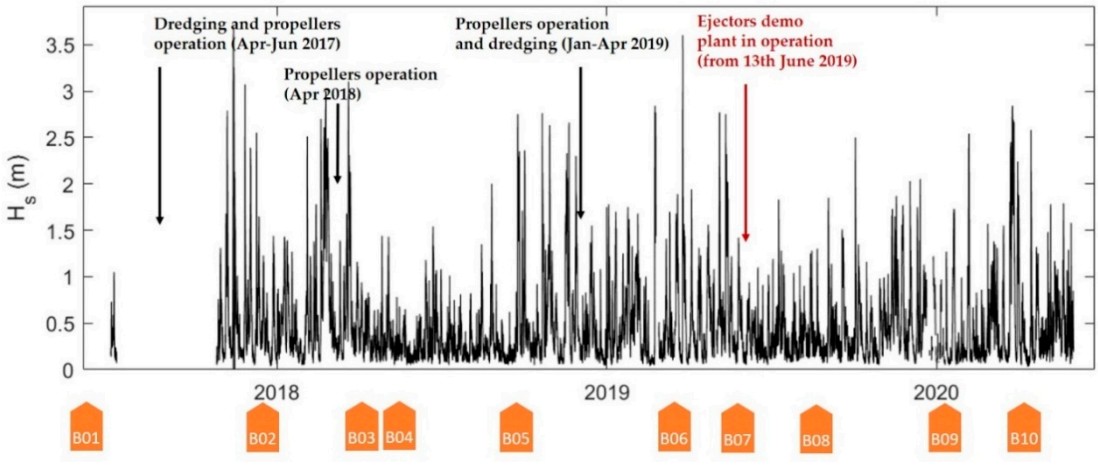

**Figure A7.** Time series of the significant wave heights in the analysed period. The date of the sediment movimentation actions and of the realized bathymetries (BXX) are indicated by arrows and tabs, respectively.

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
