# Peer review of "Effectiveness Assessment of an Innovative Ejector Plant for Port Sediment Management"

_jmse, doi:10.3390/jmse9020197_

Round 1
Reviewer 1 Report
The authors present a highly interesting study on the use of a new, innovative dredging system for port maintenance. The effectivity of the system is evaluated on the basis of a series of bathymetrical surveys over a three-year period, along with the associated wave forcing. The results are of great relevance for research and practice, yet it takes effort to derive these from the paper. For example:
- The abstract is not sufficiently informative. It describes which research activities were carried out, but does pay any attention to the outcomes of the study.
- Some of the key wording uses raises confusion. Most notably in the use of the word ‘water level’(= elevation of water surface), where bed level is meant
- The same holds for the figures. I would strongly recommend to use different color scales if different variables are shown (eg water depth versus sedimentation/erosion)
- Important background information is missing. For instance on the layout and operation of the ejector system (discharges, sediment concentrations), and how its operation relates to the meteorological conditions.
The enclosed pdf document highlights occasions where this occurs, and also provides edits / comments / suggestions on how to bring this further.
Besides, it is important to take readers along in the choices that are made. At several occasions implicit choices are made without any further clarification or justification of the rationale behind. The paper would strongly benefit from a better explanation of the rationale behind several choices.
As regard to the content, my most import comment is that in the discussion section, a systematic analysis of the sediment balance is missing. The net sedimentation in the channel is the result of the difference between wave-driven sediment import and the ejector-induced sediment removal from the area. As it stands, there are uncertainties in the sediment influx as only storms are taken into account; the paper hypothesizes sediment transport during mild conditions plays a role. Such may be the case, but I’d like to see this substantiated for instance by means of a quick CERC calculation (if only to assess the relative difference between stormy and non-stormy conditions). There may also be better ways to assess the ejector-induced sediment removal, if necessary only qualitatively and/or in relative sense. At the end, a sediment by-pass figure of 750 m3 is given; it is not clear to me how this number was determined, nor what it reflects and how it can be compared to the longer-term dredging requirements at this port.
Finally, considering the good performance of the system, would you consider alternative layouts for future applications? Eg locate the individual ejectors further away from each other (to cover a larger area), or to locate them closer to the port entrance? Is such practically feasible? Would be nice to say a few words on this in the discussion.

Author Response
Dear Editor,
Please find our revised manuscript entitled “Effectiveness assessment of an innovative ejectors plant for port sediment management”, by Marco Pellegrini et al. We have carefully addressed the remarks that were brought up by the editor and by the reviewers of our original manuscript. Below, we have listed our responses to the comments. We appreciate the positive evaluation and the constructive comments and suggestions. These have helped us to incorporate improvements in the revised version of the manuscript.
Kind regards, also on behalf of my co-authors,
Renata Archetti

Reviewer 2 Report
This manuscript sheds light on the effect of ejectors demo plant at Cervia’s port to guarantee the navigability that undoubtedly is an important issue to improve coastal management in this area. Nevertheless, there are some points that should be addressed. Please, find them below organized by section:
Introduction
Page 2- Please, check if the ejectors demo plant has been in operation to April or to September because:
Lines 62-63: A demo plant has been realized in Cervia (Italy) and has been in operation from June 2019 to September 2020
Lines 82-83: Then, the operation period of the ejectors demo plant (June 2019-April 2020)
Description of the ejector demo plant
Pages 2- 3: Lines 88-98: It’s necessary to rewrite this paragraph because it is equal that the paragraph that appears in page 17 of reference [13]. Furthermore, it’s suggested to include the length of docks at the beginning and after the different modifications.
Page 3: Line 109: The site presents an annual net longshore sediment transport directed from North to South. It’s suggested to include the value of the annual net longshore sediment transport. Furthermore, it’s recommended to include some information about cross-shore sediment transport in the area.
Page 3. It’s recommended to include some information about peak wave period in the area.
Materials and methods
Page 7 – Line 196. A digital hydrographic ultrasound system with narrow emission cone. The model of the system might be included.
Page 7 – Line 197. Has Municipality of Cervia provided the bathymetric data (i.e., longitude, latitude and depth) to create the digital elevation model or has Municipality of Cervia provided the models? There is no information about how bathymetries have been created (i.e., software used, details about interpolation (method, output cell size)) and this information should be included.
Page 8 – Line 217. It’s recommended to include the version of QGIS used.
Page 8 – Line 218: The base level is assumed at -7 m only for computing purposes. It’s not clear the reason to use the value -7 m. Why is -7 m instead other value?
Page 9 – Line 235. It’s necessary to include the reference for the energy-based classification of the sea storm that appears in Table 2.
Pages 9 – 11 Lines 235-265. The identified sea storms events in the period 2017-2020 are a result, and thus, it’s recommended to move to Results and Discussion section.
Results and discussion
It appears that it was not a good idea to merge result and discussion section. It’s recommended a direct comparison of findings discussed in the perspective of a worldwide literature.
Pages 11-12: Lines 269-270: where the navigation channel presented a draft around -4 m, for 100 m of width m in length. This cannot be verified in Figure 11 because this figure doesn’t have a scale. For that reason, it’s important to follow the recommendations given for figures (see Figures section).
Page 12: Lines 288-289: On May 2018 a new propellers operation was needed since the port entrance was substantially closed. Here, it might be referred the value of water depth at port entrance that prevent the navigation.
Conclusions
Page 18: Line 438: the first industrial size demo plant at the port entrance of the Marina of Cervia (Italy). This is the first time that Marina of Cervia has been referred because previously, it was only mentioned Cervia.
FIGURES
Figure 1:
This Figure is equal that Figure 2 in reference [14]. Therefore, it’s necessary to change it. Please, find below the recommendations:
Left panel: It’s highly recommended to add coordinates, north arrow, scale and the words ‘Italy, Adriatic Sea, …’ in Figure. Furthermore, it might be added the position of the NAUSICAA buoy in Figure.
Right panel: It’s recommended to add coordinates, north arrow and scale.
Figure 4:
- Sketch of the ejector. This sketch is equal that the sketch that appears at reference [14]. It’s necessary to indicate this in Figure caption.
Figure 6:
The sketch of the P&ID of the pumping plant also appears at reference [13]. It’s necessary to indicate this in Figure caption.
Figure 8:
It’s recommended to add coordinates, north arrow and scale.
Figure 9:
At the axis Y, please change ‘seastorms’ by ‘sea storms’.
Figures 11, 12, 13, 14, 15, 16, 17:
Maps must have coordinates, north arrow and scale according to map’s definition. Therefore, it’s necessary to add these elements in these Figures. Furthermore, maps might have a basemap (e.g., aerial picture from Figure 1). It’s recommended to change the color of mooring points of the ejectors’ inlet and outlet pipelines because red color is being used for depth.
It’s highly recommended to use the same scale for all maps related to water depth [Figure 16 b) and Figure 17 have different scales]. As the minimum water level at the port entrance to guarantee navigability is -2.5 meters, it’s recommended to consider this when classes would be defined. As water depth is negative, it’s recommended to write negative values in the water depth scale.
TABLES
Table 3. Sea storm events in the period 2018-2020, together with the list of bathymetry surveys and sediment movimentation at the port entrance: peak wave significant height (Hs), mean (Tm) and peak (Tp) wave period, mean wave direction (MWD), compass sector, storm duration (dur), storm energy 241 (E), energetic class, sea level at the Hs instant and max sea level. [a] Sea level data not available; [b] Nausicaa data not available
Please, change ‘2018’ by ‘2017’ because storms have been identified since 2017. Please, change ‘peak wave significant height’ from ‘maximum significant wave height’.
In the table, it’s recommended to include the units between parenthesis as it was done in Table 2.
APPENDIX A
It’s suggested to include a plot with the significant wave height serie where the date of the bathymetries might be pointed with an arrow.
REFERENCES
There are references with name and surname whereas there are others with initial name and surname. Please, see the following examples:
[2] Pedro Magaña, Miguel Á. Reyes-Merlo, Ángela Tintoré, Carmen Zarzuelo and Miguel Ortega-Sánchez. An 471 Integrated GIS Methodology to Assess the Impact of Engineering Maintenance Activities: A Case Study of 472 Dredging Projects. J. Mar. Sci. Eng. 2020, 8(3), 186; https://doi.org/10.3390/jmse8030186
[7] Bianchini A., Cento F., Guzzini A., Pellegrini M. & Saccani C. (2019) - Sediment management in coastal 485 infrastructures: Techno-economic and environmental impact assessment of alternative technologies to 486 dredging. Journal of Environmental Management, 248: 1 - 17.
It's recommended to always use the same format.
Author Response
Dear Reviewer,
Please find our revised manuscript entitled “Effectiveness assessment of an innovative ejectors plant for port sediment management”, by Marco Pellegrini et al. We have carefully addressed the remarks that were brought up by the editor and by the reviewers of our original manuscript. Below, we have listed our responses to the comments. We appreciate the positive evaluation and the constructive comments and suggestions. These have helped us to incorporate improvements in the revised version of the manuscript.
Kind regards, also on behalf of my co-authors,
Renata Archetti
